# ViDiT-Q: Efficient and Accurate Quantization of Diffusion Transformers for Image and Video Generation

**Tianchen Zhao[12], Tongcheng Fang[12], Haofeng Huang[1], Rui Wan[1], Widyadewi Soedarmadji[1], Enshu Liu[1]**
**Shiyao Li[1], Zinan Lin[3], Guohao Dai[24], Shengen Yan[2], Huazhong Yang[1], Xuefei Ning[1*], Yu Wang[1*]**
[1] Tsinghua University, [2] Infinigence AI, [3] Microsoft, [4] Shanghai Jiaotong University

## Abstract

Diffusion transformers have demonstrated remarkable performance in visual generation tasks, such as generating realistic images or videos based on textual instructions. However, larger model sizes and multi-frame processing for video generation lead to increased computational and memory costs, posing challenges for practical deployment on edge devices. Post-Training Quantization (PTQ) is an effective method for reducing memory costs and computational complexity. When quantizing diffusion transformers, we find that existing quantization methods face challenges when applied to text-to-image and video tasks. To address these challenges, we begin by systematically analyzing the source of quantization error and conclude with the unique challenges posed by DiT quantization. Accordingly, we design an improved quantization scheme: ViDiT-Q (**V**ideo & **I**mage **Di**ffusion **T**ransformer **Q**uantization), tailored specifically for DiT models. We validate the effectiveness of ViDiT-Q across a variety of text-to-image and video models, achieving W8A8 and W4A8 with negligible degradation in visual quality and metrics. Additionally, we implement efficient GPU kernels to achieve practical 2-2.5x memory saving and a 1.4-1.7x end-to-end latency speedup.

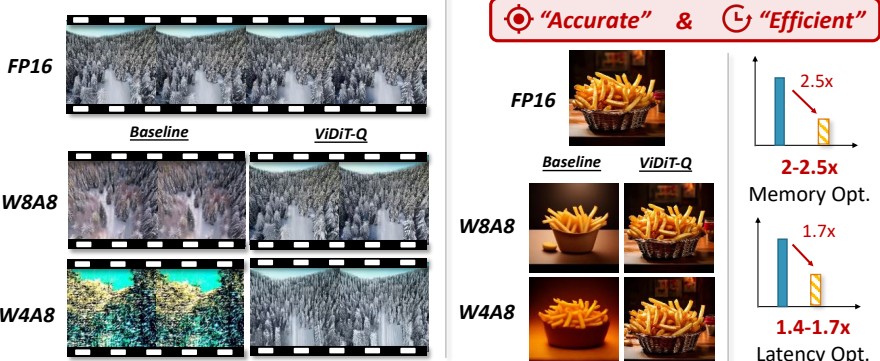

Figure 1: **ViDiT-Q** addresses the challenges that existing quantization methods face in text-to-image and video generation. It achieves quantization with negligible performance loss, delivering 2-2.5x memory savings and 1.4-1.7x latency reduction.

## 1 Introduction

Diffusion Transformers (DiTs) Peebles & Xie (2023) and video generation tasks Singer et al. (2022) have garnered significant research interest since the impressive performance of SORA OpenAI (2024). However, the increasing model size poses challenges for application and deployment on

---
*Corresponding Authors

edge devices. In the realm of video generation, processing multiple frames imposes a significant burden on both memory and computation. For example, the OpenSORA HPC-AI (2024) model consumes over 10 GB of GPU memory to generate a single $512\times512$ resolution video with only 16 frames, taking about 50 seconds on an Nvidia A100 GPU.

Model quantization Jacob et al. (2018) has proven to be an effective compression method, and is compatible with diffusion efficiency improvement techniques from other perspectives such as efficient sampling Liu et al. (2023a;b); Ni et al. (2024); Yuan et al. (2024a) and caching Ma et al. (2023); Zhang et al. (2024a; 2025a); Zou et al. (2024) . By compressing high bit-width floating-point (FP) data into lower bit-width integers, the computational and memory costs can be effectively reduced. The quantization of DiT models remain under-explored. While some prior studies Wu et al. (2024); Chen et al. (2024) explores DiT quantization for class-conditioned generation, we empirically obeserve challenges when applying them to more challenging text to image and video generation tasks with larger-scaled model (as seen in Fig. 1). Another line of recent literature Zhang et al. (2024c;b; 2025b); Yuan et al. (2024b); Pu et al. (2024) focus on optimizing the attention computation, while we focus on quantizing the linear layers.

To address this challenge, we begin by analyzing the sources of quantization error and conclude the primary issue stems from improperly large quantization range caused by high data variation within quantization groups. Next, we investigate the unique challenges in the specific application. For DiT models, we observe significant variation in multiple dimensions. For visual generation task, we find that merely reducing quantization error is insufficient to preserve the multi-faceted generation quality, such as textual alignment Wu et al. (2021) and temporal consistency Liu et al. (2023d).

In light of the above findings, we further investigate the reason for the failure of existing methods and introduce corresponding modifications. First, to handle data variation in multiple dimensions, we carefully examine the limitations of quantization grouping of existing methods from the perspectives of both algorithm performance and hardware efficiency, and highlight the need for fine-grained and dynamic quantization parameters. Second, in response to the unique time-varying channel imbalance problem, we analyze the shortcomings of existing scaling and rotation based channel balancing techniques, and design a "static-dynamic" channel balancing technique that combines the strengths of both approaches. Finally, to preserve multiple aspects of generation quality under lower bitwidth, we introduce a metric-decoupled mixed precision scheme, which "decouples" the effects of quantization across different dimensions for sensitivity analysis.

We summarize our contributions as follows:

1. We conduct extensive analysis and identify the major source of quantization error and unique challenges for quantizing the DiT model and visual generation task.

2. We design improved quantization scheme ViDiT-Q, tailored for DiT models, containing techniques accordingly to address these challenges.

3. We validate the effectiveness of ViDiT-Q on extensive DiT models for both image and video generation, and further implement efficient GPU kernels to achieve practical hardware savings and acceleration.

## 2 RELATED WORKS

### 2.1 DIFFUSION TRANSFORMERS FOR IMAGE AND VIDEO GENERATION

Diffusion Transformers (DiTs), which employ Transformers Vaswani et al. (2017) to replace the CNN-based diffusion backbones (U-Net Ronneberger et al. (2015)) in prior research Rombach et al. (2022), have achieved remarkable performance in visual generation. **Image Generation:** DiT Peebles & Xie (2023) and UViT Bao et al. (2023) pioneer the use of transformers as diffusion backbones. PixArt-$\alpha$ Chen et al. (2023) explores text-to-image generation with DiTs. **Video Generation:** Early video generation models Ho et al. (2022b;a); Guo et al. (2023) mainly adopted CNN backbones. Latte Ma et al. (2024) pioneer the use of transformers for text-to-video generation. The success of SORA OpenAI (2024) inspire the development of video diffusion transformers such as OpenSORA HPC-AI (2024). Both high-resolution image generation and multi-frame video generation add to hardware costs, necessitating efficiency improvements.

## 2.2 IMAGE AND VIDEO GENERATION EVALUATION METRICS

Visual generation can be evaluated from multiple aspects, and many metrics are introduced accordingly. **Image Metrics:** FID Heusel et al. (2017) and IS Salimans et al. (2016) are two commonly adopted metrics for measuring the Inception network feature difference between generated and reference images for quality and fidelity assessment. ClipScore Hessel et al. (2021) evaluates how well the generated image follows the prompt instruction (text-image alignment), while ImageReward Xu et al. (2023), HPS Wu et al. (2023b) incorporates human preference by collecting actual user data to train the reward model. **Video Metrics:** FVD extends the feature-based metric FID to the video domain. CLIPSIM Wu et al. (2021) estimates the similarity between video and text instructions. CLIP-temp Esser et al. (2023) measures the semantic similarity between video frames. Flow-score is proposed as part of the video evaluation benchmark EvalCrafter Liu et al. (2023c) to assess motion quality. EvalCrafter also adopts DOVER Wu et al. (2023a) for video quality assessment. These **metrics from multiple aspects should be considered** when evaluating the effect of quantization.

## 2.3 MODEL QUANTIZATION

Post Training Quantization (PTQ) has proven to be an efficient and effective model compression method Nagel et al. (2021). **Diffusion Model:** Focusing on the unique timestep dimension, prior research Q-Diffusion Li et al. (2023) and PTQ4DM Shang et al. (2023) collects timestep-wise activation data to determine quantizaiton parameters. **Transformer:** Prior research made significant progress in quantizing transformers for both ViTs Liu et al. (2021) and language models Yao et al. (2022). One major focus is addressing the channel imbalance issue. SmoothQuant Xiao et al. (2024) introduces channel-wise scaling to balance the difficulty of weight and activation quantization, while Quarot Ashkboos et al. (2024) employs orthogonal matrix rotations to distribute values more evenly across channels. **DiT:** Q-DiT Chen et al. (2024) tackles channel-wise imbalance by assigning different quantization parameters to different channels. PTQ4DiT Wu et al. (2024) addresses time-varying channel imbalance by designing a fixed channel balance mask that fits all timesteps. While these methods improve quantization from various angles, **directly applying them to the more challenging task of text-to-image/video generation in DiT models results in notable performance degradation**. In Sec. 4, we thoroughly discuss their limitations and propose novel techniques to overcome these challenges.

## 3 PRELIMINARY ANALYSIS

### 3.1 QUANTIZATION ERROR ANALYSIS

Consider the quantization problem as seeking the optimal quantization strategy to minimize the difference between the quantized model and the floating-point model. An usual approach is to surrogate this task into minimzing the layer-wise quantization error for weight $W$ and activation $X$:

$$\min \mathcal{L}_{\text{task}}(f_{FP}, f_q) \quad \Rightarrow \quad \min_{W_q, X_q} \sum_{l}^{L} \left( \|W^{(l)} - Q(W^{(l)})\|_2^2 + \|X^{(l)} - Q(X^{(l)})\|_2^2 \right), \quad (1)$$

where $f_{FP}$, $f_q$ denotes the network with $L$ layers. The $W_q$, $X_q$ represents quantized weight and activation. The weight and activation are quantized within each group $G$ (e.g., tensor-wise, channel-wise). The quantization process approximates the full-precision $x$ with integer $x_{\text{int}}$ and quantization parameters (scaling factor $s$, zero point $z$): $x \approx \hat{x} = s(x_{\text{int}} - z)$. The elements within certain group of size $g$, represented by vector $x \in R^g$ shares the same quantization parameters ($s$ and $z$). The quantization operator $Q$ with $b$ bits is described as:

$$x_{\text{int}} = Q(x; s, z, b) = \text{clamp}\left(\left\lfloor \frac{x}{s} \right\rceil + z, 0, 2^b - 1\right). \quad (2)$$

The function $\text{clamp}(x; a, c)$ clamps the values into range $[a, c]$, the $\lfloor \cdot \rceil$ is the round-to-nearest operator. As discussed in prior literature Nagel et al. (2021), the quantization error mainly consist of two parts, the clamping error and the rounding error. They act as a trade-off, the clamping error could

be reduced with larger scaling $s$, however, this in turn increases the rounding error, which lies in range $[-\frac{s}{2}, \frac{s}{2}]$. In the minmax quantization scheme adopted by most recent literature and deployment tools, the scaling $s = (\max(x) - \min(x))/(2^b - 1)$ are chosen to set the quantization range in $[\max(x), \min(x)]$, which avoids the clipping error. Therefore, **the major source of the quantization error arises from the rounding error with large $s$ when large data variation exists within the group**. For example, when the group size is large (i.e., tensor-wise), the range are determined by a small portion of large values, making the quantization range unnecessarily large for the majority of elements, thus resulting in larger rounding error. Recent literature Chee et al. (2024) introduces the idea of "incoherence processing" echoes this finding. The data group $x \in R^g$ is defined to be $\mu$-coherent if: $\max(x) \le \mu ||W||_F / \sqrt{g}$, where $|| \cdot ||_F$ indicates the Frobenius norm, and $g$ is the number of elements. The data group with higher incoherence are harder to quantize, since the largest element is an outlier relative the average magnitude. Additional incoherence processing to ensure balanced data distirbution within group is essential to reduce the quantization error.

### 3.2 Unique Challenges for DiTs and Visual Generation

**Challenges for DiT model:** As mentioned above, data variation within group incur large quantization error. We conduct comprehensive analysis for DiT data distribution, and discover that DiT model witness **high data variation in multiple levels** as presented in Fig. 2:

- **Token-wise Variation:** We observe notable variation between the visual tokens. Specifically, for video DiTs, the variations exist both along the spatial and temporal dimension.

- **Condition-wise Variation:** For conditional generation, the classifier-free-guidance Ho (2022) (CFG) conducts two separate forwards with and without the control signal (often implemented with batch of 2). We observe notable difference between the conditional part (red square) and the unconditional part (blue square).

- **Timestep-wise Variation:** Diffusion method iterates the model for multiple timesteps. We observe notable variation in activation for the same layer across timesteps.

- **Channel-wise Variation:** For both the weight and activation, we witness significant difference across different channels. Specifically, the activation channel variation demonstrate time-varying characteristics.

**Challenges for visual generation task:** As described in eq. (2), the minimization of quantization error is often adopted as the proxy task for quantization. However, for visual generation, the generation quality could be evaluated from multiple perspectives (e.g., aesthetic, alignment). **Simply regularizing the absolute error may not be sufficient for assessing the quantization's effect on visual generation.** (discussed in more details in Sec. 4.3) For video generation, more aspects related to the temporal dimension should be included (e.g., temporal consistency, temporal flickering).

## 4 ViDiT-Q: Quantization Scheme tailored for DiTs

As presented in Fig. 2, to address the aforementioned challenges, we design **ViDiT-Q**. Firstly, we highlight the importance of choosing fine-grained and dynamic quantization parameters to avoid data variation in large group. (Sec. 4.1). Secondly, we design a static-dynamic channel balance technique to handle the unique time-varying channel-wise data variation within group (Sec. 4.2). Finally, considering the quantization's effect of multiple aspects on generation quality, we design metric decoupled mixed precision method to preserve performance under lower bitwidths (Sec. 4.3).

### 4.1 Fine-grained Grouping and Dynamic Quantization

As discussed in Sec. 3.1, high data variation within the quantization group (i.e., high incoherence) is a major source of quantization error. Adopting coarse-grained quantization grouping with larger group size is more likely to include data with higher variation. For instance, in tensor-wise quantization groupings, as used in prior research Wu et al. (2024), the group contains all tokens, leading to high variation (as shown in Fig. 2). This suggests that **finer groupings should be used as long as they do not impede efficient hardware implementation**. In hardware implementations, the data

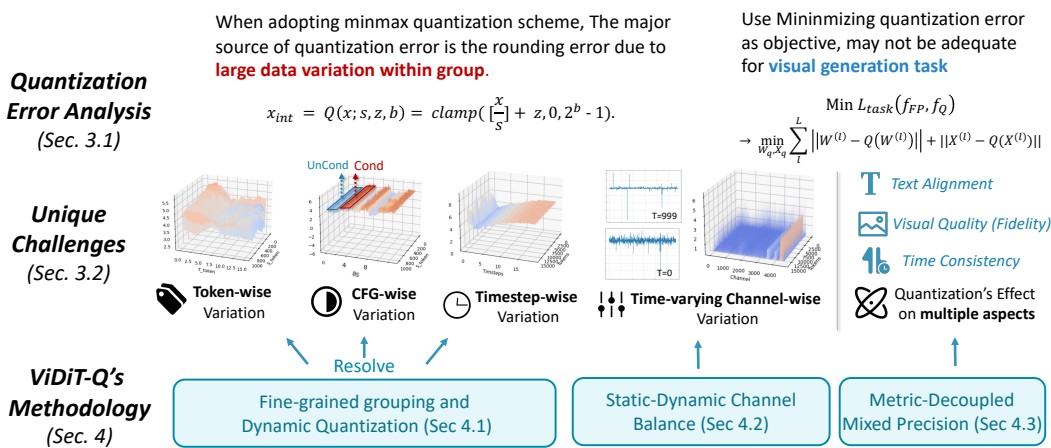

Figure 2: **The overall framework of ViDiT-Q.** We begin by analyzing the sources of quantization error and identifying the unique challenges faced by DiT models and visual generation tasks. Correspondingly, we develop specialized techniques to address these challenges.

summed together should share the same quantization parameters (i.e., belong to the same quantization group) to avoid the overhead of casting integer values to floating-point for summation. In transformer quantization, where the majority of computation occurs in the Linear layers, summation occurs along the input-channel dimension of the weights and activations. Therefore, despite the "channel-wise" activation quantization in Q-DiT Chen et al. (2024) enhances performance, it brings difficulty in hardware acceleration. Differently, we adopt the hardware-friendly "channel-wise" and "token-wise" quantization groupings for weights and activation. This approach compresses the group size for activation quantization to the number of channels, introducing negligible overhead compared to coarse-grained groupings, and is supported by mainstream inference frameworks Zhao et al. (2024b); Lin et al. (2024).

For diffusion model, two additional dimensions, "condition-wise" and "timestep-wise" variation are introduced. **Using static quantization parameters across all timesteps and conditions results in equivalent larger group sizes with larger data variation.** For example, PTQ4DiT Wu et al. (2024) adopts the tensor-wise static activation quantization grouping, and fails to hanlde the high variation in the token, timestep dimensions. To address timestep-wise variation, previous methods So et al. (2024) adopt timestep-wise static quantization parameters. However, the determination of these quantization parameters are costly (requires iterative training) and face difficulties when generalizing across solvers. In contrast, we propose using "dynamic" quantization parameters, which are computed online and naturally adapt to varying timesteps and conditions. This approach acts as the upper bound of algorithm performance for resolving the timestep-wise variation issue. The additional hardware cost is negligible, as it only requires determining the max and min of the data group and can be fused with previous operations to further minimize overhead. More detailed profiling and analysis are presented in Sec. 5.3.

## 4.2 STATIC-DYNAMIC CHANNEL BALANCING

As mentioned above, by incorporating fine-grained grouping and dynamic quantization, the data group is reduced to a vector with $C$ channels. Consequently, **reducing data variation within the group (i.e., channel balancing) is crucial for minimizing quantization error**. As illustrated in Fig. 2, channel-wise data variation is evident in transformer models. Specifically, for DiTs, the degree of channel imbalance varies significantly across timesteps. Existing channel-scaling or rotation based techniques struggle with the unique "time-varying channel imbalance". We investigate the reasons for their failure and design a specialized "static-dynamic" channel balance technique.

Scaling based methods Xiao et al. (2024) introduce a per-channel balancing mask $s \in \mathcal{R}^{C_i}$. By dividing the activation with $s$ and multiplying $s$ with weights, it shifts the quantization difficulty from activation to weights, and vice versa. The mask $s$ could be calculated as follows:

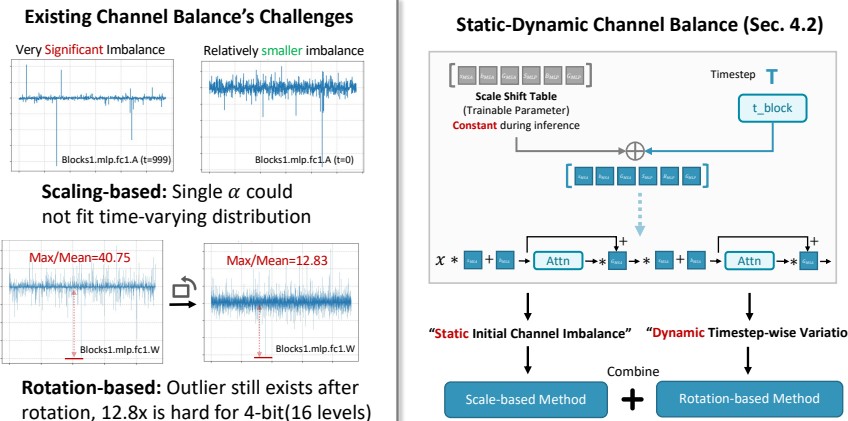

Figure 3: **The illustration of static-dynamic channel balancing.** Left: the limitations of existing rotation and scaling based channel balancing techniques. Right: the reason for time-varying imbalance could be decomposed into the static and the dynamic part.

$$Y = (X\text{diag}(s)^{-1}) \cdot ((\text{diag}(s)W)) = \hat{X} \cdot \hat{W}; \quad s_i = \max(|X_i|)^{\alpha}/\max(|W_i|)^{1-\alpha}, \quad (3)$$

where $X, Y, W$ represents the input activation , output activation, and weights. The $s$ is a channel-wise balancing mask, $\alpha$ is a hyperparameter. Channel balancing could effectively alleviate the input channel-wise variation. However, we empirically discover that it is sensitive to $\alpha$ choices. **For different timesteps, the degree of activation channel imbalance changes, suitable $\alpha$ also changes**. Employing the same $\alpha$ for earlier stages may shift too much difficulty from weights to activations, harming the activation quantization, and vice versa for latter stages. Introducing multiple $\alpha$s for different timesteps can resolve this issue. However, it necessitates different versions of weights for various timesteps. Optimizing for the optimal $\alpha$ cross all timesteps is alos challenging.

Rotation based methods Ashkboos et al. (2024); Liu et al. (2024) introduces an orthogonal rotation matrix $Q$, such that $QQ^T = I$ and $|Q| = 1$. Multiplying the matrix $Q$ on the left and right of the data preserves computational invariance $Y = XW^T = (XQ)(Q^TW)$. The rotation matrix makes the data values more evenly distributed along channels. Quantizing the rotated matrix $XQ$ with less incoherence could reduce quantization error. The rotation based method requires no parameter tuning and naturally adjust to varying degree of channel imbalance across timestep. However, as seen in Fig. 3, **some channels are still prominently larger than others after the rotation**.

To overcome these limitations, we analyze the data distribution of DiT models and discover that the time-varying channel imbalance phenomenon arises from the "feature modulation" that aggregates the timestep embedding with the feature (as shown in Fig. 3). This phenomenon can be decomposed into two parts: the "static" initial activation distribution orginating from the pretrained "scale shift table" and the "dynamic" variation introduced by the time embedding. Inspired by these findings, we propose combining scaling and rotation-based channel balancing methods to leverage the strengths of both. The scaling-based method addresses the "static" channel imbalance at the initial denoising stage, avoiding the need for multiple $\alpha$s for varying distributions. The rotation-based method is then utilized to address the "dynamic" varying distribution. Since the scaling method has already alleviated extreme channel imbalance, the rotation method ensures a balanced distribution.

### 4.3 METRIC DECOUPLED MIXED PRECISION DESIGN

The aforementioned techniques can effectively reduce the "incoherence" of data distribution, thereby decreasing quantization error. However, we still observe notable quality degradation with lower bit-widths (W4). Upon investigating the reasons for this issue, we find that some layers, despite exhibiting relatively low quantization error, can significantly impact overall quantization. This suggests that layers have varying quantization sensitivity, and quantization can be "bottlenecked" by certain highly sensitive layers. This aligns with the discussions in Sec. 3.1 and Sec. 3.2. Merely

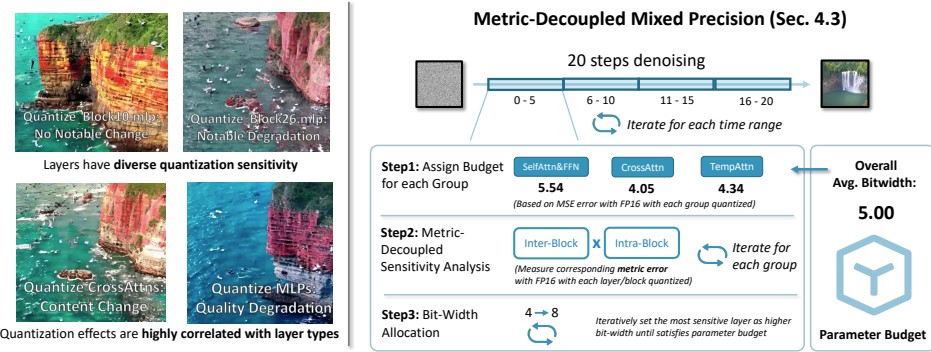

Figure 4: **The illustration of metric decouple mixed precision.** Left: The findings of layer sensitivity and quantization's effect. Right: the overview of metric decoupled mixed precision.

minimizing quantization error may not be sufficient, especially for visual generation tasks. **Quantization's effects on multiple aspects of generation quality should be considered.**

To address the "bottleneck" phenomenon, a straightforward solution is to assign higher bit-widths to "protect" these sensitive layers. The main challenge in mixed precision allocation lies in accurately identifying these sensitive layers. Prior literature Yang et al. (2023) measures sensitivity by quantizing specific layers and calculating the Mean Squared Error (MSE) with the floating-point output. However, MSE alone does not accurately reflect overall generation performance. Consistent with previous studies Sui et al. (2024), we discover that MSE tends to overemphasize content changes at the expense of visual quality degradation and temporal consistency.

Inspired by prior research Zhao et al. (2024a), we find that **quantization's impact on various aspects of generation quality is strongly correlated with layer types.** Specifically, cross-attention layers contribute significantly more to "content change" than other layers. In contrast, "visual quality" is primarily influenced by spatial attention and feedforward network (FFN) layers, while "temporal consistency" is chiefly affected by temporal attention layers. Given these diverse impacts on generation, the quantization effects of these layers should not be directly compared.

Building on this finding, we propose to **"disentangle" the mixed influences of quantization on multiple aspects** and develop a metric-decoupled mixed precision method as presented in Fig. 4. To account for sensitivity variation across timesteps, we divide the denoising process into four equal ranges and conduct sensitivity analysis for them respectively. Given a target bit-width budget, we categorize layers into three groups and allocate the budget based on the MSE error between FP16 generation and each group when quantized. For each group, we use specific metrics (VQA. ClipScore, and FlowScore) as sensitivity measures for the layers within that group. Finally, we iteratively assign higher bit-widths to the most sensitive layers within each group until reaches the budget.

## 5 EXPERIMENTS

### 5.1 IMPLEMENTATION DETAILS AND EXPERIMENTAL SETTINGS

**Video Genration Evaluation Settings:** We apply ViDiT-Q to OpenSORA HPC-AI (2024), the videos are generated with 100-steps DDIM with CFG scale of 4.0. The mixed precision is only adopted for the challenging W4A8. The evaluation contains two settings. (1) **Benchmark suite:** We evaluate the quantized model on VBench Huang et al. (2023) to provide comprehensive results. Following prior research Ren et al. (2024), we select 8 major dimensions from Vbench. (2) **Multi-aspects metrics:** We select representative metrics, and measure them on OpenSORA prompt sets. Following EvalCrafter Liu et al. (2023c), we select *CLIPSIM* and *CLIP-Temp* to measure the text-video alignment and temporal semantic consistency, and DOVER Wu et al. (2023a)'s video quality assessment (*VQA*) metrics to evaluate the generation quality from aesthetic (high-level) and technical (low-level) perspectives, *Flow-score* and *Temporal Flickering* are used for evaluating the temporal consistency. We also present results on the UCF-101Soomro et al. (2012), adopting FVD Unterthiner et al. (2019) as the metric for OpenSORA and Latte Ma et al. (2024) in the Appendix Sec. D.1.

Table 1: **Performance of ViDiT-Q text-to-video generation on VBench evaluation benchmark suite.** The bit-width "16" represents FP16 without quantization. We omit some baselines that fails to produce readable content under W4A8. The mixed precision are applied for ViDiT-Q W4A8.

| Method | Bit-width (W/A) | Imaging Quality | Aesthetic Quality | Motion Smooth. | Dynamic Degree | BG. Consist. | Subject Consist. | Scene Consist. | Overall Consist. |
|---|---|---|---|---|---|---|---|---|---|
| - | 16/16 | 63.68 | 57.12 | 96.28 | 56.94 | 96.13 | 90.28 | 39.61 | 26.21 |
| Q-Diffusion | 8/8 | 60.38 | 55.15 | 94.44 | 68.05 | 94.17 | 87.74 | 36.62 | 25.66 |
| Q-DiT | 8/8 | 60.35 | 55.80 | 93.64 | 68.05 | 94.70 | 86.94 | 32.34 | 26.09 |
| PTQ4DiT | 8/8 | 56.88 | 55.53 | 95.89 | 63.88 | 96.02 | 91.26 | 34.52 | 25.32 |
| SmoothQuant | 8/8 | 62.22 | 55.90 | 95.96 | 68.05 | 94.17 | 87.71 | 36.66 | 25.66 |
| Quarot | 8/8 | 60.14 | 53.21 | 94.98 | 66.21 | 95.03 | 85.35 | 35.65 | 25.43 |
| ViDiT-Q | 8/8 | 63.48 | 56.95 | 96.14 | 61.11 | 95.84 | 90.24 | 38.22 | 26.06 |
| Q-DiT | 4/8 | 23.30 | 29.61 | 97.89 | 4.166 | 97.02 | 91.51 | 0.00 | 4.985 |
| PTQ4DiT | 4/8 | 37.97 | 31.15 | 92.56 | 9.722 | 98.18 | 93.59 | 3.561 | 11.46 |
| SmoothQuant | 4/8 | 46.98 | 44.38 | 94.59 | 21.67 | 94.36 | 82.79 | 26.41 | 18.25 |
| Quarot | 4/8 | 44.25 | 43.78 | 92.57 | 66.21 | 94.25 | 84.55 | 28.43 | 18.43 |
| ViDiT-Q | 4/8 | 61.07 | 55.37 | 95.69 | 58.33 | 95.23 | 88.72 | 36.19 | 25.94 |

| Method | Bit-width (W/A) | CLIPSIM | CLIP-Temp | VQA-Aesthetic | VQA-Technical | Δ Flow Score. (↓) |
|---|---|---|---|---|---|---|
| - | 16/16 | 0.1797 | 0.9988 | 63.40 | 50.46 | - |
| Q-Diffusion | 8/8 | 0.1781 | 0.9987 | 51.68 | 38.27 | 0.328 |
| Q-DiT | 8/8 | 0.1788 | 0.9977 | 61.03 | 34.97 | 0.473 |
| PTQ4DiT | 8/8 | 0.1836 | 0.9991 | 54.56 | 53.33 | 0.440 |
| SmoothQuant | 8/8 | 0.1951 | 0.9986 | 59.78 | 51.53 | 0.331 |
| Quarot | 8/8 | 0.1949 | 0.9976 | 58.73 | 52.28 | 0.215 |
| ViDiT-Q | 8/8 | 0.1950 | 0.9991 | 60.70 | 54.64 | 0.089 |
| Q-DiT | 6/6 | 0.1710 | 0.9943 | 11.04 | 1.869 | 41.10 |
| PTQ4DiT | 6/6 | 0.1799 | 0.9976 | 59.97 | 43.89 | 0.997 |
| SmoothQuant | 6/6 | 0.1807 | 0.9985 | 56.45 | 48.21 | 29.26 |
| Quarot | 6/6 | 0.1820 | 0.9975 | 61.47 | 53.06 | 0.146 |
| ViDiT-Q | 6/6 | 0.1791 | 0.9984 | 64.45 | 51.58 | 0.625 |
| Q-DiT | 4/8 | 0.1687 | 0.9833 | 0.007 | 0.018 | 3.013 |
| PTQ4DiT | 4/8 | 0.1735 | 0.9973 | 2.210 | 0.318 | 0.108 |
| SmoothQuant | 4/8 | 0.1832 | 0.9983 | 31.96 | 22.85 | 0.415 |
| Quarot | 4/8 | 0.1817 | 0.9965 | 47.36 | 33.13 | 0.326 |
| ViDiT-Q | 4/8 | 0.1809 | 0.9989 | 60.62 | 49.38 | 0.153 |

*FP16*

*ViDiT-Q*

*Q-DiT*

*PTQ4DiT*

Figure 5: **Performance of text-to-video generation on OpenSORA prompt set.** Left: The comparison of generation quality for different quantization methods under different bitwidths. The Q-diffusion for W6A6 and W4A8 are omitted since it fails to generate readable content. Right: Visualization of generated videos for DiT quantization methods under W4A8.

**Image Evaluation Settings:** We apply ViDiT-Q to PixArt-$\alpha$ model, the images are generated with 20-steps DPM-solver with CFG scale of 4.5. No mixed precision is adopted for W4A8. We choose *FID* Heusel et al. (2017) for fidelity evaluation, *Clipscore* Hessel et al. (2021) for text-image alignment, and *ImageReward* Xu et al. (2023) for human preference. These metrics are measured on the first 1024 prompts on COCO annotations.

**Hardware Implemention Settings**: We implement efficient quantized GEMM CUDA kernels for practical resource savings. Following SmoothQuant Xiao et al. (2024), the scaling-based channel balance factors are fused into the previous layer. Kernel fusion is also adopted by integrating the quantization operation and Hadamard transformation into the previous LayerNorm, GeLU, and residual operations, thereby minimizing the quantization overhead. We measure the latency and memory savings on the Nvidia A100 GPU using CUDA12.1, the memory usage is measured with the PyTorch Memory Management APIs PyTorch (2023), and the latency is profiled with Nsight tools NVIDIA. The profiling is conducted with batch size of 1, and 20 denoising steps.

| Method | Bit-width (W/A) | FID(↓) | CLIP(↑) | IR(↑) |
|---|---|---|---|---|
| - | 16/16 | 73.34 | 0.258 | 0.901 |
| Q-Diffusion | 8/8 | 96.54 | 0.239 | 0.186 |
|  | 4/8 | 91.95 | 0.228 | -0.224 |
| Q-DiT | 8/8 | 73.60 | 0.256 | 0.854 |
|  | 4/8 | 475.8 | 0.127 | -2.277 |
| PTQ4DiT | 8/8 | 127.9 | 0.217 | -1.216 |
|  | 4/8 | 171.9 | 0.177 | -2.064 |
| ViDiT-Q | 8/8 | 75.61 | 0.259 | 0.917 |
|  | 4/8 | 74.33 | 0.257 | 0.887 |

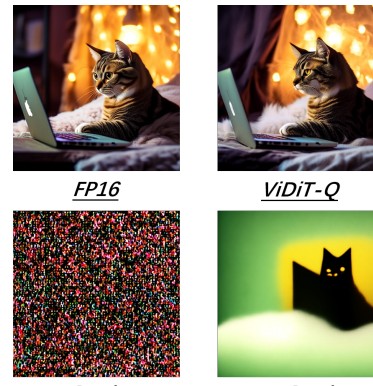

Figure 6: **Performance of ViDiT-Q text-to-image generation on COCO.** Left: The metric scores of PixArt-$\alpha$ quantization. Right: Generated images comparison of W4A8 quantization.

## 5.2 MAIN RESULTS

**Text-to-video generation on VBench and OpenSORA prompt set:** As presented in Tab. 1 and Fig. 5, existing diffusion quantization methods (e.g., Q-Diffusion) designed for U-Net-based models incur notable quality degradation (VQA in Fig. 5, and mutliple aspects in Tab. 1) even at W8A8. These methods often fail to generate readable content (resulting in blank images or noise) at lower bit-widths, which are omitted in the table. The primary reason for their failure is the use of coarse-grained and static quantization parameters. The baseline DiT quantization methods (Q-DiT, PTQ4DiT) achieve acceptable performance at W8A8 and W6A6. However, for W4A8, they fail to manage channel imbalance, producing unreadable content, as shown in the right part of Fig. 5. For Q-DiT, its grouping mechanism struggles to handle the large output channel variation under W4. For PTQ4DiT, the use of fixed channel balancing scaling is insufficient to manage the large variation across timesteps. Language model quantization techniques (e.g., SmoothQuant, Quarot) also perform comparably at W8A8 and W6A6. However, significant degradation is observed at the more challenging W4A8, highlighting the importance of improved channel balancing.

**Text-to-image generation on COCO:** Similar to video generation, as seen in Fig. 6, existing quantization schemes, which employ fine-grained static quantization parameters (Q-Diffusion and PTQ4DM), encounter challenges even with W8A8 configurations. These difficulties arise from the improper handling of activation data variation across multiple dimensions. While Q-DiT achieves satisfactory performance under W8A8, it struggles under W4A8 due to output channel-wise imbalance. In contrast, ViDiT-Q consistently maintains performance across all bitwidths.

## 5.3 HARDWARE RESOURCE SAVINGS

**Memory footprint reduction.** Fig. 7 (a) shows the GPU memory usage of ViDiT-Q and the FP16 baseline. ViDiT-Q can reduce the memory from two aspects: (1) Weight quantization reduces the allocated memory for storing model weights. (2) Activation quantization reduces allocated memory to store intermediate activations. Combining the two benefits, ViDiT-Q can effectively reduce the peak memory footprint by $1.99\times$ for W8A8. Due to the adoption of mixed precision, the memory savings under W4A8 are slightly less than the theoretical value, achieving a $2.42\times$ reduction.

**Latency speedup.** We present the latency speedup in Fig. 7-(b). Replacing FP16 layers with our efficient INT8 kernels achieves approximately $2\times$ acceleration. Taking into account the unquantizable layers (e.g., norms, non-linears, attention) and the overhead from quantization (FP to INT conversion, channel balancing scaling, and rotations), the overall speedup is $1.71\times$. We also compare our quantization method with a standard baseline ("Naive W8A8") that lacks dynamic and fine-grained quantization as well as channel balancing techniques. As shown, the incorporation of our methods significantly improves generation quality while introducing only minimal hardware overhead (from $1.73\times$ to $1.71\times$). For W4A8, since current DiT computation exhibits a "compute-bound" characteristic, the W4A8 CUDA kernel primarily saves memory without enhancing efficiency. Due to the use of mixed precision, the overall latency speedup is smaller ($1.38\times$) compared to W8A8.

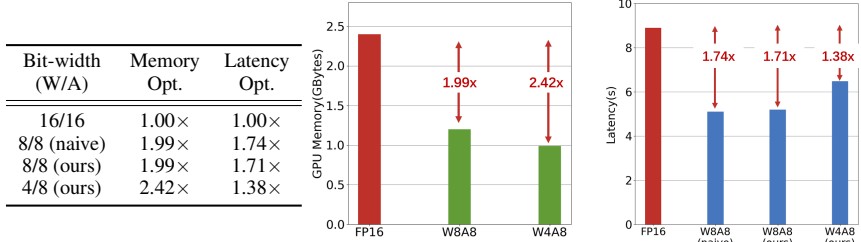

| Bit-width (W/A) | Memory Opt. | Latency Opt. |
|---|---|---|
| 16/16 | $1.00\times$ | $1.00\times$ |
| 8/8 (naive) | $1.99\times$ | $1.74\times$ |
| 8/8 (ours) | $1.99\times$ | $1.71\times$ |
| 4/8 (ours) | $2.42\times$ | $1.38\times$ |

Figure 7: **The illustration of ViDiT-Q's hardware resource savings.** The table and figures present memory savings and end-to-end latency speedup of ViDiT-Q and naive quantization scheme.

Table 2: **Ablation studies of ViDiT-Q techniques.** The comparison of OpenSORA W4A8 performance by gradually incorporating ViDiT-Q techniques.

| Methods | | | Bit-width | CLIPSIM | CLIP-Temp | VQA- | VQA- | Δ Flow |
|---|---|---|---|---|---|---|---|---|
| Quant Params | Channel Balance | Mixed Precision | (W/A) | | | Aesthetic | Technical | Score. |
| - | - | - | 16/16 | 0.180 | 0.998 | 64.198 | 51.904 | - |
| Static & Tensor-wise | - | - | 4/8 | 0.201 | 0.997 | 0.178 | 0.086 | 0.603 |
| Dynamic & Token-wise | - | - | 4/8 | 0.196 | 0.998 | 32.217 | 10.994 | 0.109 |
| Dynamic & Token-wise | Scaling-based | - | 4/8 | 0.191 | 0.999 | 31.963 | 22.847 | 0.415 |
| Dynamic & Token-wise | Rotation-based | - | 4/8 | 0.181 | 0.999 | 47.356 | 33.128 | 0.326 |
| Dynamic & Token-wise | Static-Dynamic | - | 4/8 | 0.181 | 0.999 | 60.216 | 42.257 | 0.151 |
| Dynamic & Token-wise | Static-Dynamic | MSE-based | 4/8 | 0.179 | 0.999 | 53.335 | 38.729 | 0.258 |
| Dynamic & Token-wise | Static-Dynamic | Metric Decoupled | 4/8 | 0.199 | 0.999 | 60.616 | 49.383 | 0.334 |

## 5.4 ABLATION STUDIES

We present ablation studies that gradually incorporate ViDiT-Q's techniques for W4A8 quantization, as shown in Tab. 2. **Effectiveness of fine-grained grouping and dynamic quantization parameter:** Replacing static, tensor-wise quantization parameters with dynamic, token-wise ones significantly improves generation, transforming it from near failure (with VQA scores close to zero) to producing readable content. **Effectiveness of static-dynamic channel balancing:** The scaling and rotation-based channel balancing technique results in notable quality degradation, whereas static-dynamic balancing improves generation quality to a level comparable to FP. **Effectiveness of mixed precision:** Applying metric-decoupled mixed precision further enhances generation quality, while MSE-based mixed precision negatively impacts performance.

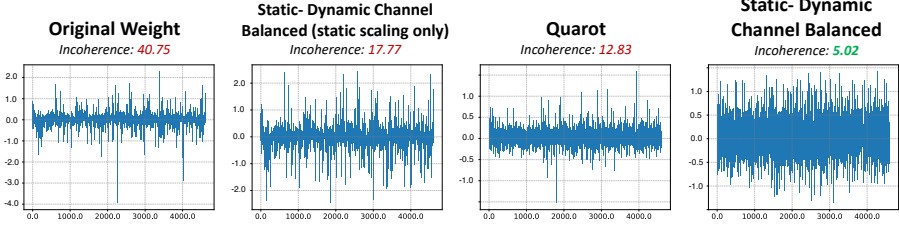

Figure 8: **The channel distribution of block1.mlp.fc1 weights with different channel balancing techniques.** Directly adopting Quarot still results in relatively large incoherence. However, implementing a static-dynamic channel balance reduces incoherence to an acceptable level..

## 6 CONCLUSION AND LIMITATIONS

We design ViDiT-Q, a quantization method that addresses unique challenges of DiTs. It achieves W4A8 quantization with minimal performance degradation for popular video and image generation models. We further implement CUDA kernels to achieve 2-2.5$\times$ memory savings, and 1.4-1.7$\times$ latency speedup. Despite achieving good performance, the mixed precision design is still worth polishing, and lower activation bit-width is essential for fully utilizing the acceleration potential of 4-bit weight. We aim to address these issues and further improve ViDiT-Q.

## REPRODUCIBILITY STATEMENT

All findings presented in this paper are fully reproducible. We have provided anonymized code in the supplementary materials, along with the videos shown in the figures. Detailed information about our experiments, including hyperparameters, training protocols, and evaluation methods, is available in the Experiments section. We are confident that, with the provided resources, readers will be able to reproduce all of the results presented.

## ACKNOWLEDGEMENT

This work was supported by National Natural Science Foundation of China (No. 62325405, 62104128, U19B2019, U21B2031, 61832007, 62204164), Tsinghua EE Xilinx AI Research Fund, and Beijing National Research Center for Information Science and Technology (BNRist). We thank for all the support from InfinigenceAI.

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

# A ADDITIONAL EXPERIMENTAL DETAILS

## A.1 MOTIVATION FOR QUANTIZING LINEAR LAYERS ONLY

In Sec. 4, we mention that the linear layers accounts for the most of the computation. So we focus on quantizing the linear layers and leave the attention computation unquantized. We elaborate on the reason for this focus here. In Fig. 9, we visualize the detailed latency breakdown for an STDiT model block. The 'attention computation' includes the matrix multiplication for query and key embedding to generate the attention map, and the multiplication of the attention map with the value embedding. The QKV linear mapping and the projection after attention aggregation are not included, as these are linear layers that can be quantized. As shown, when utilizing FlashAttention, the latency cost of attention computation accounts for only 14.3% of the overall latency. Additionally, FlashAttention minimizes the activation memory usage for storing the attention map. Therefore, we focus on the primary cost: the linear layers. We quantize all linear layers except for the "t embedding", "y embedding" and "final layer", they appear at the start or end of the model, and have smaller channel sizes. They account for only negligible amount of computation ($< 1/1000$ overall latency), therefore we maintain them as FP16.

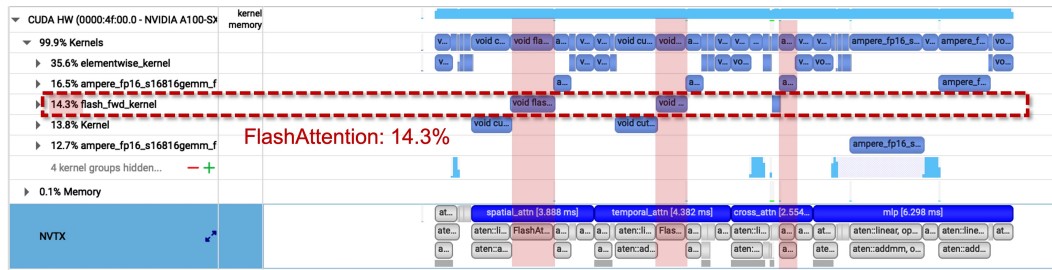

Figure 9: **The latency comparison of linear layers and attention computation.** When utilizing FlashAttention, the attention computation only takes up a small portion (14.3%) of the latency.

## A.2 IMPLEMENTION DETAILS FOR BASELINE METHODS

**Q-Diffusion:** We follow the official open-sourced code and collect timestep-wise activation as calibration data, and conduct optimization for the scaling factor "delta", and the AdaRound parameter "alpha". **PTQ4DiT:** We reimplement the "$\rho$-guided" saliency correction. Following the original paper, we adopt static tensor-wise activation parameters. **Q-DiT:** Following the original paper, we adopt channel-wise quantization grouping for both the weight and activation. The group size is determined with the evolutionary algorithm search. **SmoothQuant:** We adopt the same formulation of channel scaling mask $s$, and leverage grid search to determine the optimal $\alpha$. **QuaRot:** We adopt the same Hadamard matrix transformation as the original paper. Specifically, since we donot quantize the attention QK matrix multiplication, we only apply rotation to linear layers.

# B DETAILED DESCRIPTION OF EVALUATION METRICS

## B.1 BENCHMARK SUITE

Following VBench Huang et al. (2023), our benchmark suite encompasses three key dimensions.
(1) **Frame-wise Quality** assesses the quality of each individual frame without taking temporal quality into concern.

- **Aesthetic Quality** evaluates the artistic and beauty value perceived by humans towards each video frame.
- **Imaging Quality** assesses distortion (e.g., over-exposure, noise) presented in the generated frames

(2) **Temporal Quality** assesses the cross-frame temporal consistency and dynamics.

- **Subject Consistency** assesses whether appearance of subjects in the video remain consistent throughout the whole video.
- **Background Consistency** evaluates the temporal consistency of the background scenes.
- **Motion Smoothness** evaluates whether the motion in the generated video is smooth and follows the physical law of the real world.
- **Dynamic Degree** evaluates the degree of dynamics by calculating average optical flow on each video frame.

(3) **Semantics** evaluates the video's adherence to the text prompt given by the user. consistency.

- **Scene Consistency** evaluates whether the video is consistent with the intended scene described by the text prompt.
- **Overall Consistency** reflects both semantics and style consistency of the video.

We utilize three prompt sets provided by official github repository of VBench. We generate one video for each prompt for evaluation.

- **subject_consistency.txt:** include 72 prompts, used to evaluate subject consistency, dynamic degree and motion smoothness.
- **overall_consistency.txt:** include 93 prompts, used to evaluate overall consistency, aesthetic quality and imaging quality.
- **scene.txt:** include 86 prompts, used to evaluate scene and background consistency.

### B.2 SELECTED METRICS

**FVD and FVD-FP16:** FVD measures the similarity between the distributions of features extracted from real and generated videos. We employ one randomly selected video per label from the UCF-101 dataset (101 videos in total) as the reference ground-truth videos for FVD evaluation. We follow Blattmann et al. (2023) to use a pretrained I3D model to extract features from the videos. Lower FVD scores indicate higher quality and more realistic video generation. However, due to relatively smaller video size (e.g. 101 videos in our case), employing FVD to evaluate video generation models faces several limitations. Small sample size cannot adequately represent either the diversity of the entire dataset or the complexity and nuances of video generation, leading to inaccurate and unstable results. To mitigate limitations above, we propose an enhanced metric, FVD-FP16, for assessing the semantic loss in videos generated by quantized models relative to those produced by pre-quantized models. Specifically, we utilize 101 videos generated by the FP16 model as ground-truth reference videos. The FVD-FP16 has significantly higher correlation with human perception.

**CLIPSIM and CLIP-temp:** The CLIPSIM and CLIP-temp metrics are computed using implementation from EvalCrafter Liu et al. (2023c). For CLIPSIM, We use the CLIP-VIT-B/32 model Radford et al. (2021) to compute the image-text CLIP similarity for all frames in the generated videos and report the averaged results. The metric quantify the discrepancy between input text prompts and generated videos. For CLIP-temp, we use the same model to compute the CLIP similarity of each two consecutive frames of the generated videos and then get the averages on each two frames. The metric indicates semantics consistency of generated videos.

**DOVER's VQA:** We employ the Dover Wu et al. (2023a) method to assess generated video quality in terms of aesthetics and technicality. The technical rating(VQA-T) measures common distortions like noise, blur and over-exposure. The aesthetic rating(VQA-A) reflects aesthetic aspects such as the layout, the richness and harmony of colors, the photo-realism, naturalness, and artistic quality of the frames.

**Flow Score:** We employ flow score proposed by Liu et al. (2023c) to measure the general motion information of the video. we use RAFT Teed & Deng (2020), to extract the dense flows of the video in every two frames. Then, we calculate the average flow on these frames to obtain the average flow score of each generated video.

**Temporal Flickering:** We utilize the temporal flickering score provided by VBench Huang et al. (2023) to measure temporal consistency at local and high-frequency details of generated videos. We calculate the average MAE(mean absolute difference) value between each frame.

## C    DETAILED ANALYSIS OF EXPERIMENTAL RESULTS

In this section, we present more detailed analysis of the experimental results in Sec. 5.

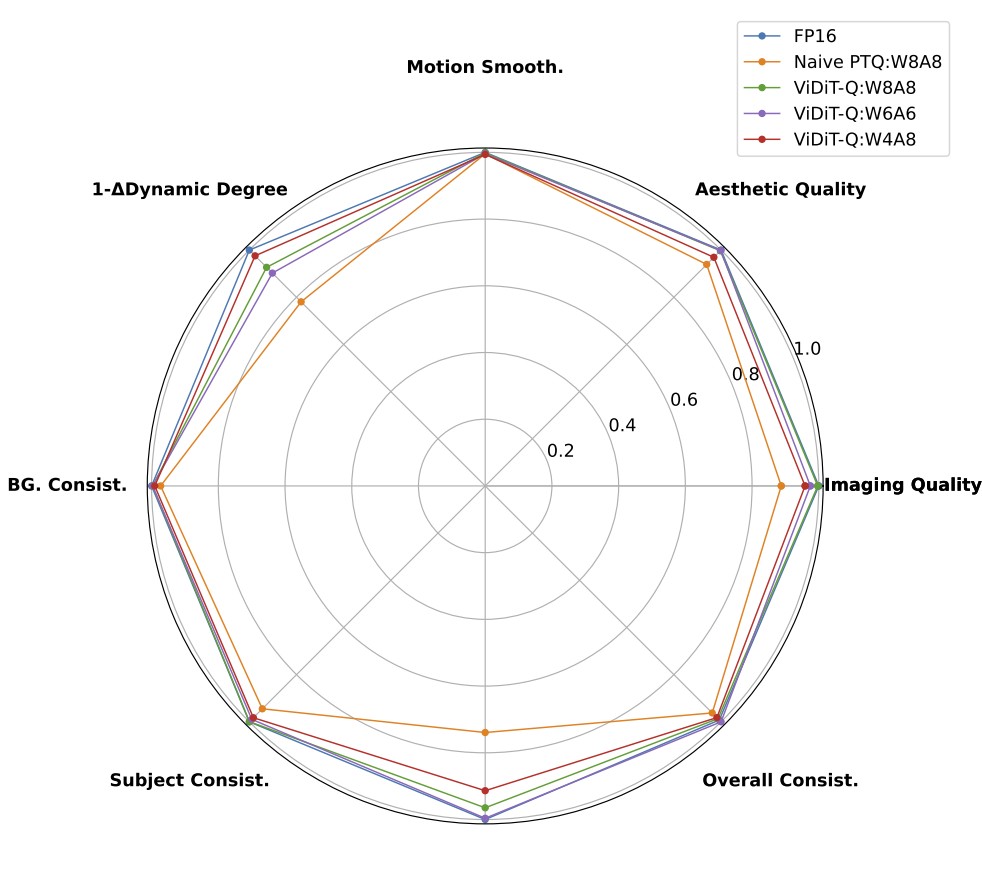

Figure 10: **The radar chart corresponding to the data presented in Table 1 from Sec. 5.2.** ViDiT-Q has a superior performance on VBench compared with the naive PTQ.

### C.1    TEXT-TO-VIDEO PERFORMANCE ON VBENCH

VBench is a comprehensive benchmark suite for video generation models, covering a wide range of dimensions, such as motion smoothness and subject consistency. The metric values of ViDiT-Q's performance on VBench is presented in Tab. 1 Sec. 5. We visualize the Radar plot of the VBench performance in Fig. 10, the metric values are normalized by the maximum value in each diemsnion. It's clearly illustrated that ViDiT-Q achieves similar performance with FP16 for all bit-widths (W8A8 , W6A6 mixed precision, W4A8 mixed precision), outperforming the Naive PTQ W8A8. We further analyze the generated video's performance from three aspects as follows:

**Dynamic Degree:**    Dynamic degree indicates the range of motion in the video, higher dynamic degree denotes more dynamic movement in the video. Lower dynamic degree denotes that the video barely moves, resembling a static image. Normally, higher dynamic degree is favored. However, in the quantization scenario, we discover that quantization often causes the generated videos to jitter and tremble. It is not favorable but results in notable dynamic degree value increase. In our

experimental setting, **too high or too low dynamic degree means degradation**. Therefore, in the radar plot, using FP16 generated videos as the ground-truth reference, we use the $(f_Q - f_{FP})/f_{FP}$ to denote "relative dynamic degree changes from FP generated videos", and use $1 - (f_Q - f_{FP})/f_{FP}$ as dynamic degree scoring in the radar plot. As illustrated Fig. 10 dynamic degree dimension, Naive PTQ W8A8's scoring ($< 0.8$) is notably lower than ViDiT-Q results. The video examples in Fig. 11 supports this finding. In Fig. 11c, the navive PTQ W8A8 generated buildings have jittering and glitches, and changes significantly across frames (ref the supplementary for the video). In contrast, both the FP16 and ViDiT-Q W8A8 generated buildings moves acutely.

**Consistency:** The consistency denotes whether some object remains consistent (does not disappear, change significantly) across frames. Vbench evaluates consistency from the subject, scene, background, and overall level. From the Radar plot, we witness ViDiT-Q also notably outperforms naive PTQ, especially in the "scene consistency" dimension ($< 0.8$). As seen in the aforementioned video example in Fig. 11c, the buildings (act as the "scene") changes significantly across frames. It violates the scene consistency and lead to lower scoring. Also, as presented in Fig. 12c, the generated bear's ear does not exist in earlier frames, and suddenly appears. This also reflects the degradation of subject consistency.

**Quality:** VBench evaluates the quality from both the aesthetic (composition and color), and imaging quality (clarity, exposure) dimension. Fig. 13 shows the example of Naive PTQ W8A8's quality degradation. The color notably turns blue, and the mountain on the left is blurred. Similar color shifting degradation is also witnessed in Fig. 12c.

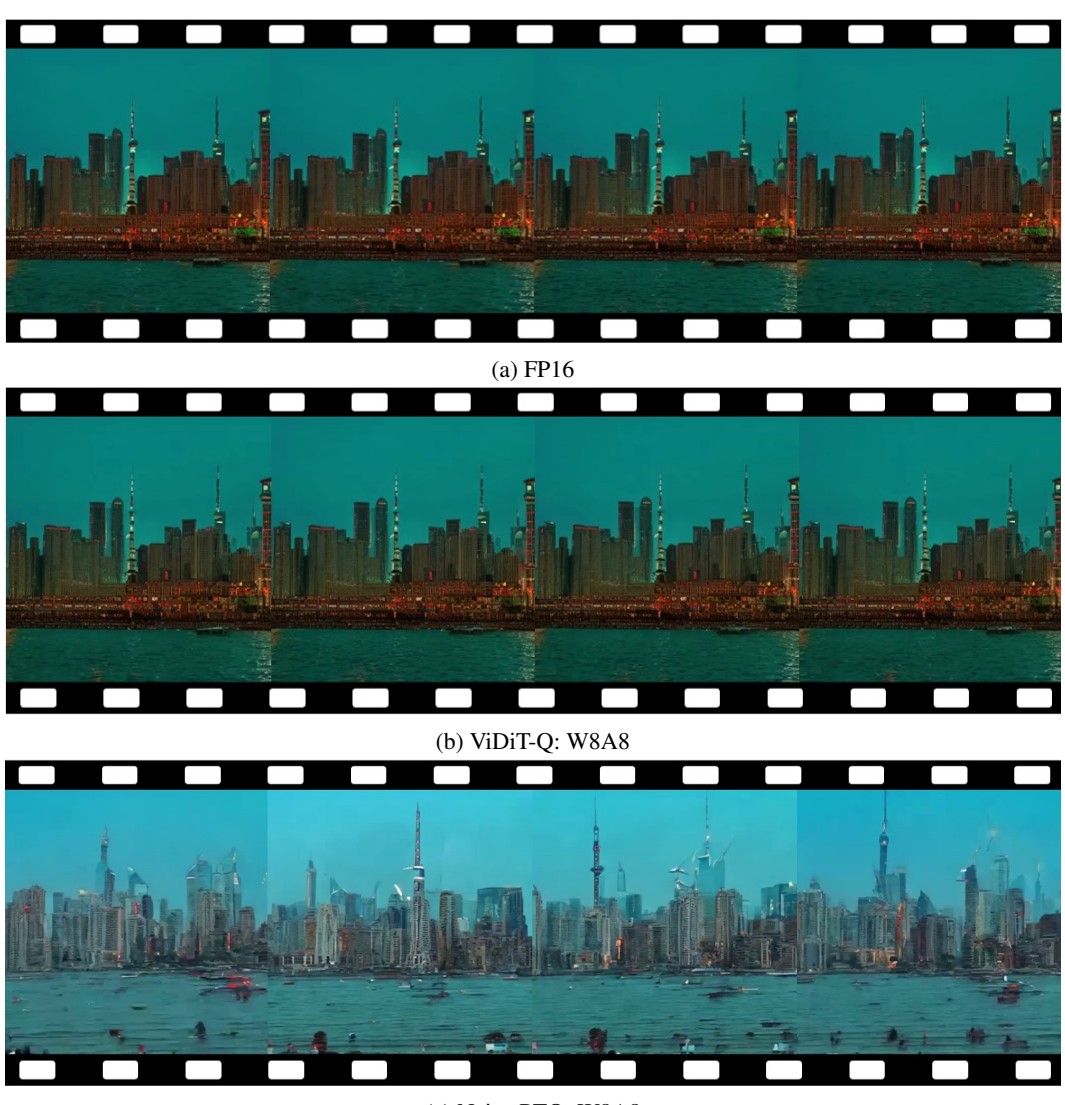

(a) FP16

(b) ViDiT-Q: W8A8

(c) Naive PTQ: W8A8

Figure 11: **The qualitative results on VBench about the ViDiT-Q's ability to maintain the dynamic degree**.

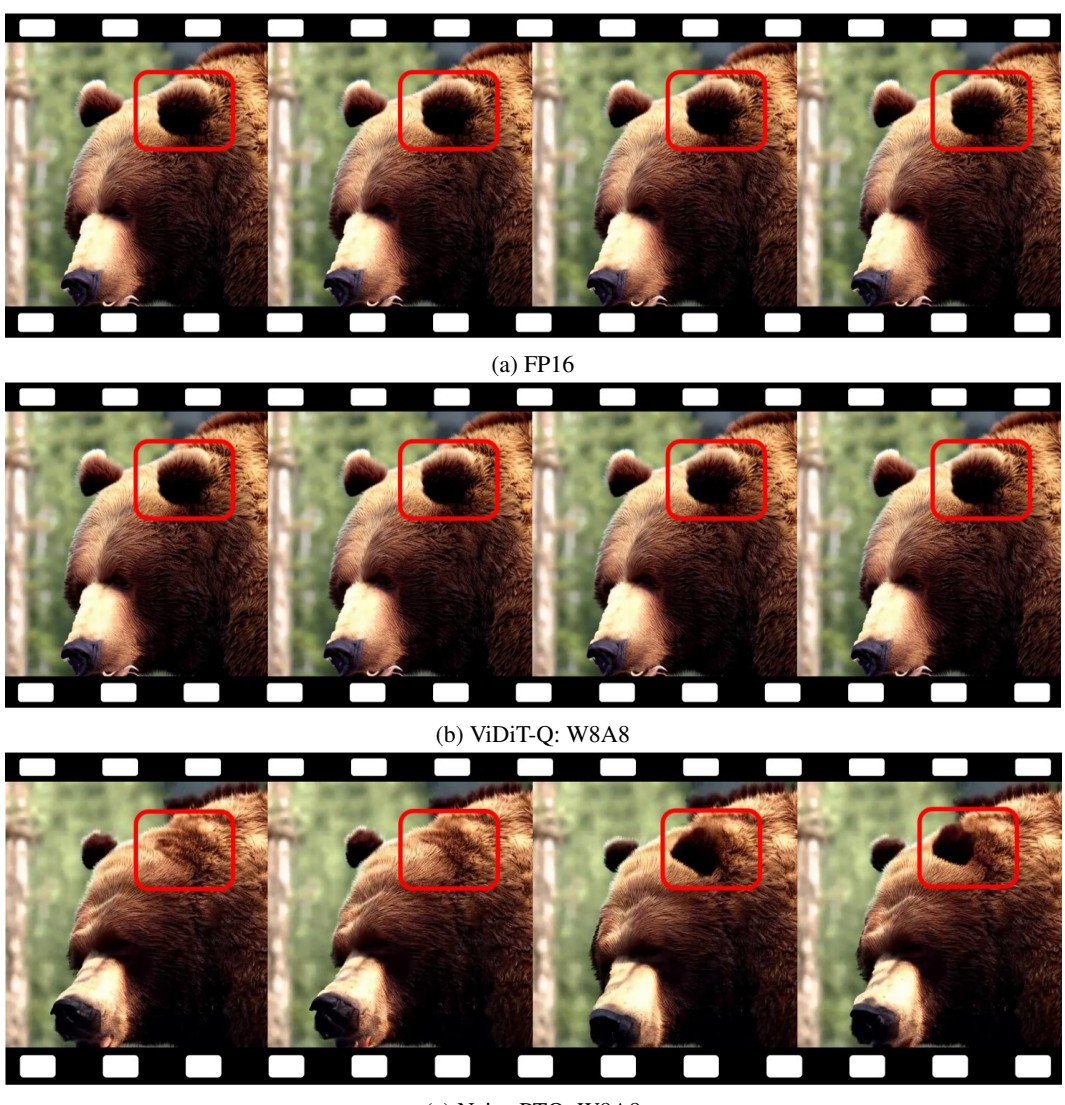

(a) FP16

(b) ViDiT-Q: W8A8

(c) Naive PTQ: W8A8

Figure 12: **The qualitative results on VBench about the ViDiT-Q's ability to maintain the consistency**.

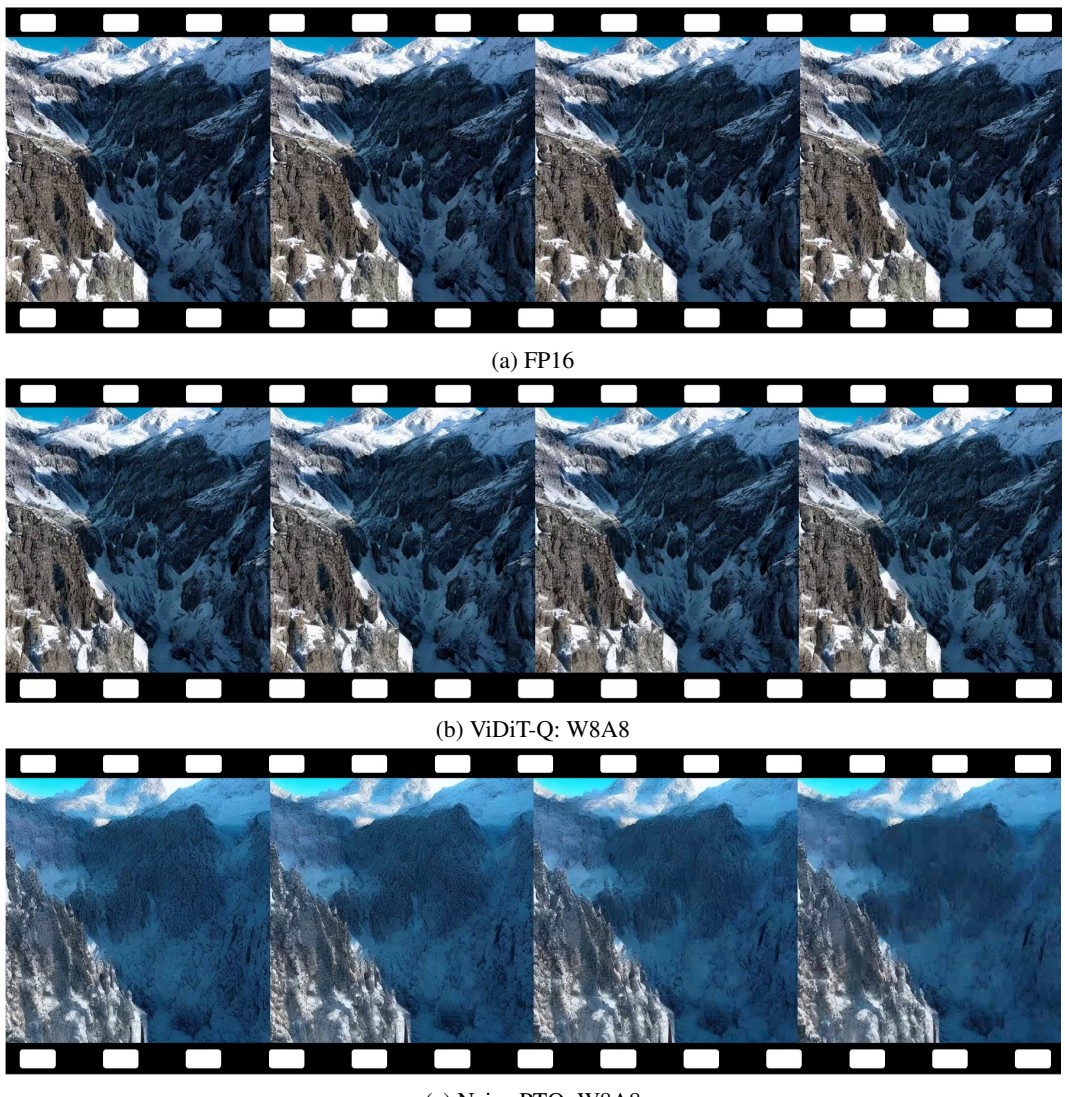

(a) FP16

(b) ViDiT-Q: W8A8

(c) Naive PTQ: W8A8

Figure 13: **The qualitative results on VBench about the ViDiT-Q's ability to maintain the image quality**.

## C.2 TEXT-TO-IMAGE GENERATION ON COCO

We present more qualitative results of generated images by baseline quantization and ViDiT-Q quantization in Fig. 14. As shown, the Naive PTQ's generated images are highly blurred. While the W8A8 images depict outlines of objects, the W4A8 images generate nearly pure noises. In contrast, ViDiT-Q generates images nearly identical to the FP16 ones, preserving both visual quality and text-image alignment.

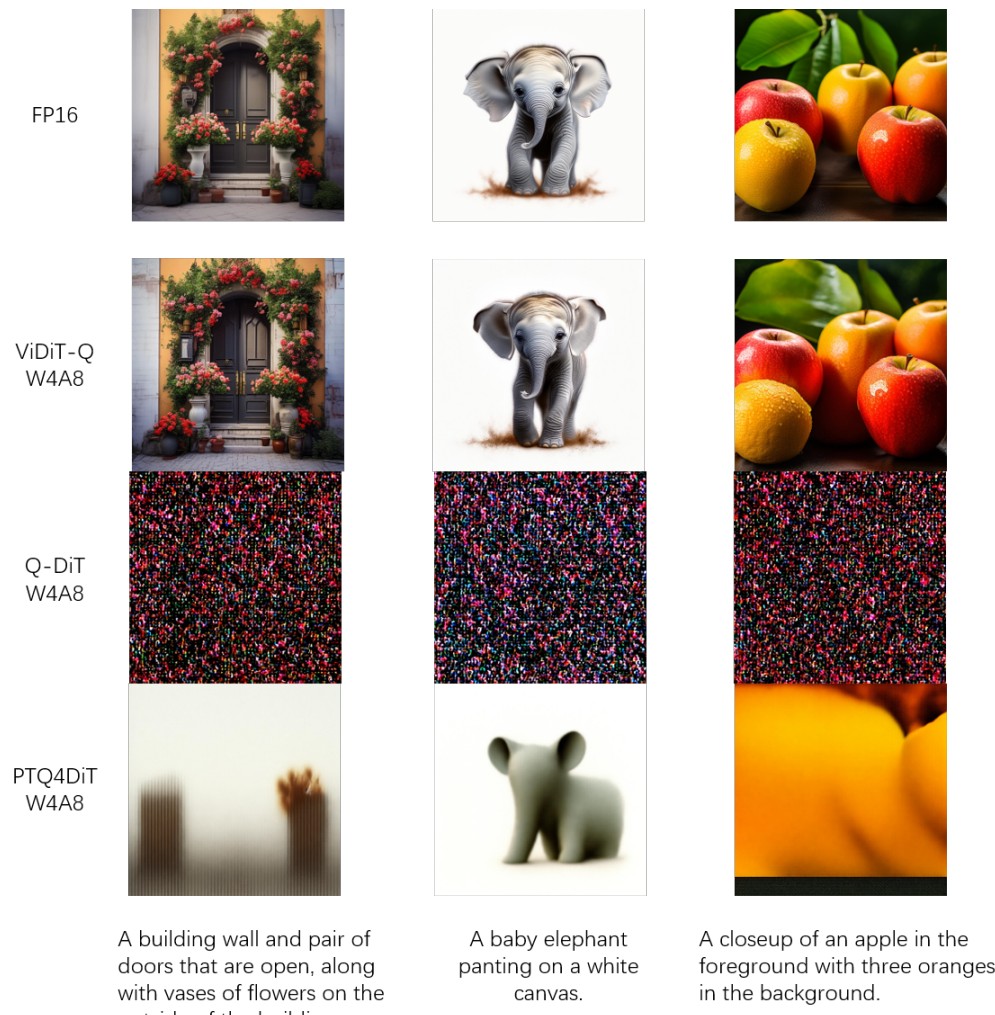

| A building wall and pair of doors that are open, along with vases of flowers on the outside of the building. | A baby elephant panting on a white canvas. | A closeup of an apple in the foreground with three oranges in the background. |

Figure 14: **Qualitative results of text-to-image generation**

## C.3 VISUALIZATION OF ABLATION STUDIES

We present the generated videos for the ablation studies in Tab. 2. As seen in Fig. 15, video quality improves from blank images to similar to the FP16 baseline. For the challenging W4A8 quantization, the baseline method generates blank images. After adding dynamic quantization, some meaningful background (deep ocean) appears, but the main object (turtle) is still missing. Channel balancing reduces color deviation (from dark blue to green-blue), but the main object remains unrecognizable and changes significantly across frames (please refer to the supplementary materials for the video). The static-dynamic channel balancing improves the consistency of the main object, but notable degradation is still observed compared to the FP16 video. Finally, with mixed precision, a similar generation quality to the FP16 baseline is achieved."

Generated Videos Example of Ablation Studies: **STDiT W4A8**

Figure 15: **Visualization of generated videos of ablation studies.**

## D ADDITIONAL EXPERIMENTAL RESULTS

### D.1 PERFORMANCE OF ViDiT-Q ON UCF-101 DATASET

In this section, we apply ViDiT-Q to both OpenSora HPC-AI (2024) Model and Latte Ma et al. (2024) and evaluate them on UCF-101 Soomro et al. (2012) dataset.

**Experimental Settings:** Similar to the settings mentioned in Sec. Sec. 5.1, We select multi-aspects metrics for more comprehensive evaluation. The commonly adopted FVD Unterthiner et al. (2019) is also provided. Specifically, due to the lack of ground-truth videos for prompt-only datasets, inspired by Tang et al. (2023), we also report FVD-FP16 which chooses the FP16 generated video as ground-truth. The above metrics are evaluated on 101 prompts (1 for each class) for UCF-101. We adopt the class-conditioned Latte model trained on UCF-101 and use the 20-steps DDIM solver with CFG scale of 7.0 for it.

**Experimental Results:** Similar to the results on VBench and OpenSORA prompt sets, for both the OpenSORA and Latte, the baseline quantization methods (Naive PTQ and Q-Diffusion) incur notable performance degradation under W8A8, and fails under W4A8. While SmoothQuant channel balance technique could achieve good performance under W8A8, it still witnesses notable degradation under W4A8. It is also worth noting that the FVD metrics are noisy when the number of videos are relatively small, and the "FVD-FP16" metric could work as an effective alternative for measuring quantization's effect.

### D.2 PERFORMANCE OF ViDiT-Q FOR SUPER RESOLUTION TASK.

ViDiT-Q addresses the core problem of reducing quantization error by analyzing the distribution of DiTs, making it highly compatible and generalizable to novel tasks that utilize DiTs. We have extended the application of ViDiT-Q to the image super-resolution task using the recent InfDiT model Yang et al. (2024). The statistics are presented in Tab. 4, and qualitative results in Fig. 16.

As could be seen, ViDiT-Q consistently maintains the performance of Inf-DiT across all bitwidths. When quantizing the model under W4A8 with ViDiT-Q, the PSNR and SSIM scores of the quantized

Table 3: **Performance of text-to-video generation on UCF-101 Dataset.**. The description of metrics is provided in Sec. 5.1, unless specified with ↓, higher metric values denote better performance.

| Model | Method | Bit-width (W/A) | FVD($\downarrow$) | FVD-FP16($\downarrow$) | CLIPSIM | CLIP-T | VQA-Aesthetic | VQA-Technical | $\Delta$ Flow Score. ($\downarrow$) | Temp. Flick. |
|-------|--------|---------|-------|----------|---------|--------|---------------|--------------|-------------|-------------|
| | - | 16/16 | 136.87 | 0.00 | 0.1996 | 0.9978 | 41.63 | 56.64 | 2.24 | 97.53 |
| | Naive PTQ | 8/8 | 154.92 | 50.72 | 0.1993 | 0.9968 | 27.52 | 35.50 | 2.61 | 97.02 |
| | Q-Diffusion | 8/8 | 144.77 | 74.97 | 0.1979 | 0.9964 | 32.88 | 44.42 | 2.50 | 96.71 |
| STDiT | SmoothQuant | 8/8 | 109.24 | 48.78 | 0.1993 | 0.9971 | 39.19 | 52.64 | 2.53 | 97.21 |
| | ViDiT-Q | 8/8 | 141.13 | 15.52 | 0.1995 | 0.9978 | 43.59 | 55.36 | 2.32 | 97.45 |
| | Naive PTQ | 4/8 | 544.34 | 637.02 | 0.1868 | 0.9982 | 0.16 | 0.13 | 1.61 | 99.90 |
| | SmoothQuant | 4/8 | 122.51 | 96.25 | 0.1960 | 0.9973 | 17.39 | 24.22 | 1.99 | 96.23 |
| | ViDiT-Q | 4/8 | 136.54 | 77.43 | 0.1978 | 0.9976 | 20.76 | 25.65 | 1.94 | 96.51 |
| | ViDiT-Q-MP | 4/8 | 129.10 | 60.13 | 0.1995 | 0.9977 | 33.98 | 47.65 | 1.8 9 | 97.57 |
| | - | 16/16 | 99.90 | 0.00 | 0.1970 | 0.9963 | 36.33 | 91.23 | 3.37 | 96.22 |
| Latte | Naive PTQ | 8/8 | 98.75 | 73.82 | 0.1981 | 0.9950 | 27.62 | 50.52 | 3.53 | 95.35 |
| | ViDiT-Q | 8/8 | 110.96 | 20.83 | 0.1959 | 0.9962 | 30.26 | 80.32 | 3.14 | 95.95 |
| | Naive PTQ | 4/8 | 183.52 | 239.08 | 0.1719 | 0.9929 | 5.62 | 0.41 | 66.06 | 65.14 |
| | ViDiT-Q | 4/8 | 95.04 | 79.11 | 0.1943 | 0.9971 | 21.76 | 32.17 | 2.84 | 95.57 |

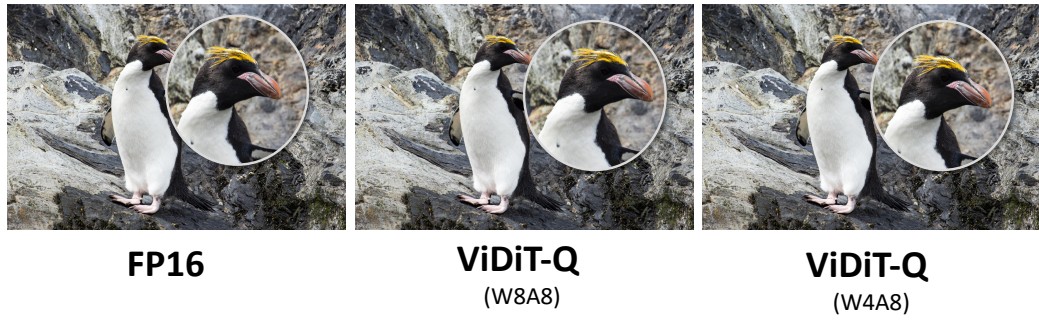

**FP16**          **ViDiT-Q**          **ViDiT-Q**
                  (W8A8)               (W4A8)

Figure 16: **Qualitative results of generated super resolution image for Inf-DiT with ViDiT-Q quantization.**

model still achieve good performance, similar to FP16. We have supplemented the implementation details as follows: We followed the settings used in the PixArt quantization experiments in our paper. For the super-resolution implementation and model evaluation, we adhered as closely as possible to the original setup. We fixed the image degradation to bicubic interpolation with 4× downsampling and conducted the experiment on the DIV2K validation dataset. Additionally, unlike other super-resolution models, Inf-DiT performs non-overlapping patch division on the downsampled images during super-resolution. This means that the original image needs to be divisible by the product of the super-resolution scale and the patch size. According to the original settings, the patch size is set to 32. Therefore, we performed center cropping on the ground truth images to ensure the image size is divisible by 128. Since the InfDiT official codebase Yang et al. (2024) did not provide detailed evaluation code, we implemented the SSIM and PSNR calculations based on the popular code repository from Saharia et al. (2021).

Table 4: **Comparison of ViDiT-Q for Inf-DiT model for image super resolution.**

| Method | PSNR | SSIM |
|--------|------|------|
| InfDiT FP16 | 25.8015 | 0.7307 |
| InfDiT (ViDiT-Q W8A8) | 25.8628 | 0.7318 |
| InfDiT (ViDiT-Q W4A8) | 26.0139 | 0.7249 |

### D.3 Efficiency Improvement on different hardware devices

We present hardware experiments on the RTX3090 and Jetson Orin Nano (a low-power embedded GPU with 7-10W power) in Fig. 17. The memory optimization for all platforms still achieves a 2x reduction, and the latency speedup varies slightly. On the RTX3090, we achieve a 1.6x latency speedup, while on the Orin, we achieve approximately 1.82x speedup. This speedup could be further improved by tuning the tiling parameters in the CUDA code, as different platforms have diverse optimal tiling parameter setups due to varying hardware resources.

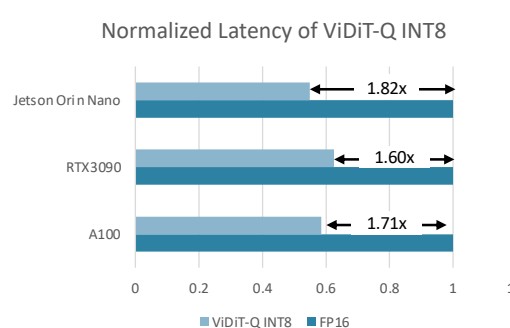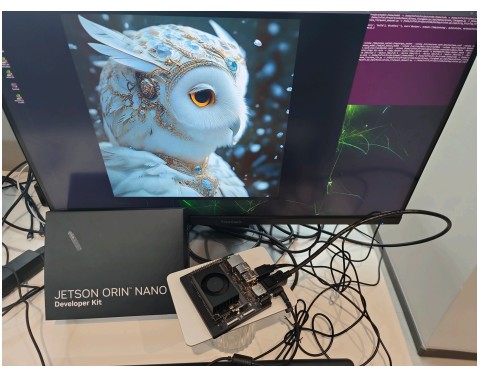

Figure 17: **Comparison of ViDiT-Q's efficiency improvement on different devices.**

### D.4 Performance of ViDiT-Q under lower bitwidth.

**Analysis of Lower Bitwidth from Hardware Perspective:** For GPUs, both the activation and weight need to be quantized into 4-bits to leverage efficient INT4 computation. In the current W4A8 implementation, the weights are quantized into 4-bits to save model size and memory cost, but they must be upcasted to 8-bit for computation with 8-bit activation. Therefore, pursuing lower bitwidths such as INT4 to fully utilize the potential of INT4 computation is definitely valid, and W2A8 could further reduce the model size and memory consumption. The W4A2 may require customized operator support, which is not currently supported by GPUs, and remains to be explored as a future direction.

We present W4A8 in the main paper as a relatively "conservative" setting to ensure negligible performance degradation. ViDiT-Q remains capable of generating images of good quality with a mixed-precision plan. We have supplemented the experiments on text-to-image generation for the Pixart-Sigma model. The statistical and qualitative examples are presented in Tab. 5 and Fig. 18 as follows. The "ViDiT-Q W4A4-MP" plan employs mixed precision without careful tuning, assigning 66.7% of the linear layers as W4A4 and the remaining rest as W8A8, resulting in an average bitwidth of approximately W5A5. The "ViDiT-Q W2A8-MP" plan assigns around 50% of the linear layers as W2A8 and the rest as W8A8, achieving an average bitwidth of approximately W5A8. 2-bit weight quantization is particularly challenging and may require quantization-aware training to preserve performance. Despite these challenges, ViDiT-Q performs well under lower bitwidth settings (W4A4 & W2A8), achieving comparable or even higher metric values compared to Q-DiT W8A8. As illustrated in the Fig. 18, even under "aggressive compression" settings, the generated images closely resemble those produced by FP models.

### D.5 Comparison with general quantization methods

we conducted experiments by adding the AdaRound Nagel et al. (2020) and BRECQ Li et al. (2021) techniques as baselines for the OpenSORA model, the implementation details of quantization techniques are set the same as Q-Diffusion, and the other implementation details are kept the same with the main paper. The results are presented in Tab. 6.

As can be seen from the table, both the AdaRound and Brecq methods experience a moderate performance drop compared to FP16, while ViDiT-Q achieves comparable results with the FP16 baseline. For the more challenging W4A8 scenario, due to the significant channel-wise variation that

Table 5: **Comparison of performance under lower bitwidths (W4A4, W2A8) for Pixart-Sigma text-to-image generation.** The "ViDiT-Q W4A4 MP" stands for utilizing the mixed precision for W4A4 quantization.

| Method (Bitwidth) | FID ($\downarrow$) | CLIP($\uparrow$) | ImageReward($\uparrow$) |
|---|---|---|---|
| FP16 | 73.34 | 0.258 | 0.901 |
| Q-DiT W8A8 | 73.60 | 0.256 | 0.854 |
| ViDiT-Q W8A8 | 75.61 | 0.259 | 0.917 |
| ViDiT-Q W4A8 | 74.33 | 0.257 | 0.887 |
| ViDiT-Q W4A4-MP | 74.56 | 0.257 | 0.861 |
| ViDiT-Q W2A8-MP | 75.32 | 0.256 | 0.843 |

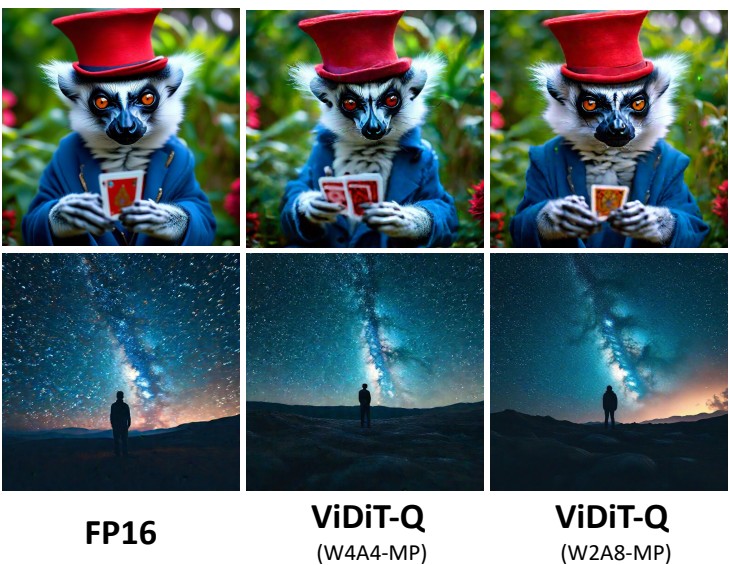

Figure 18: **Qualitative results of ViDiT-Q generated images under lower bitwidth.**

AdaRound and Brecq are not designed to handle, they fail to produce meaningful content, resulting in near-zero VQA scores. This underscores the importance of specialized techniques to address the channel imbalance problem effectively.

## D.6 COMBINATION WITH ATTENTION QUANTIZAITON METHOD

Recent attention quantization method SageAttention Zhang et al. (2024c) could reduce the cost of attention computation in DiTs through quantizing the QK into 8 bits. ViDiT-Q and SageAttention focus on different aspects of quantization. ViDiT-Q quantizes the linear layers, while SageAttention accelerates the attention QK matrix multiplication. Therefore, these two methods can be seamlessly combined to achieve better speedup. We applied SageAttention on top of the OpenSORA model as presented in Tab. 7. Since the linear layers constitute the majority of the computational cost for the model (more than 80%, as presented in Fig. 9), further introducing SageAttention will not cause notable performance degradation but could moderately improve latency. We present the algorithm performance and hardware efficiency as follows:

## E ADDITIONAL ANALYSIS

Table 6: **Comparison with general quantization methods.**

| Method | W/A | CLIPSIM | CLIP-Temp | VQA-A | VQA-T | $\Delta$ Flow Score |
|--------|-----|---------|-----------|-------|-------|---------------------|
| FP16 | - | 0.1797 | 0.9988 | 63.40 | 50.46 | 0 |
| Adaround | 8/8 | 0.1796 | 0.9983 | 52.90 | 29.84 | 0.2934 |
| Brecq | 8/8 | 0.1791 | 0.9983 | 48.27 | 31.98 | 0.3978 |
| ViDiT-Q | 8/8 | 0.1950 | 0.9991 | 60.70 | 54.64 | 0.0890 |
| Adaround | 4/8 | 0.9971 | 0.1648 | 0.272 | 0.151 | 0.4210 |
| Brecq | 4/8 | 0.1669 | 0.9963 | 0.085 | 0.077 | 0.4303 |
| ViDiT-Q | 4/8 | 0.1809 | 0.9989 | 60.62 | 49.38 | 0.1530 |

Table 7: **Comparison of efficiency when combining with SageAttention.**

| Method | Bit-width | Memory Opt. | Latency Opt. |
|--------|-----------|-------------|--------------|
| - | 16/16 | - | - |
| ViDiT-Q | 8/8 | 1.99x | 1.71x |
| ViDiT-Q + SageAttn | 8/8 | 1.99x | 1.72x |

### E.1 COMPARISON WITH BASELINE QUANTIZATION METHODOLOGY DESIGN

We present detailed comparison with existing quantization methods as follows:

**Static and coarse-grained quantization parameters:** Previous diffusion-based methods primarily focused on CNN-based model quantization (PTQ4DM Shang et al. (2023), Q-Diffusion Li et al. (2023)) adopt static and coarse-grained quantization parameters. The recent DiT-targeted quantization method PTQ4DiT Wu et al. (2024) follows this scheme. Static and coarse-grained quantization parameter determination assigns the same set of shared quantization parameters for activations across different tokens, timesteps, and conditions. As illustrated in Fig. 2, the large data variation across these dimensions incurs significant quantization errors, leading to substantial performance degradation. We also collect their performances and present them in the table below. It demonstrates that these baselines incur notable performance degradation under W8A8, and fails under W4A8.

**Channel group-wise and dynamic quantization parameters:** Q-DiT Chen et al. (2024) adopts dynamic and channel group-wise quantization parameters, where a group (64 to 128) of channels shares the same set of quantization parameters. This approach can handle channel-wise imbalance to some extent, and the "dynamic quantization" addresses variation across timesteps. However, all tokens still share the same set of quantization parameters, which is problematic for video generation models where token-wise variation is significant. This method still faces severe quality degradation. Additionally, having different quantization parameters for different channels introduces challenges for efficient CUDA implementation. As seen from the table below, the Q-DiT incurs notable performance degradation under W8A8, and fails under W4A8.

**Timestep-wise static quantization parameters:** Previous diffusion-based quantization methods that adopt static quantization parameters often determine timestep-wise quantization parameters through careful calibration (PTQ4DM Shang et al. (2023), Q-Diffusion Li et al. (2023)) and gradient-based optimization (TDQ So et al. (2024)). The existing methods for handling timestep-wise variation have the following disadvantages compared to simple dynamic quantization: (1) **The process of determining timestep-wise quantization parameters could be costly:** This process requires iterating the model multiple times for calibrating timestep-wise activation quantization, and parameter tuning like TDQ incurs additional training costs. In contrast, adopting dynamic quantization requires no overhead for determining parameters for each timestep. (2) **The determined timestep-wise quantization parameters may face challenges in generalizing across different timesteps and solvers:** Static timestep-wise quantization parameters need to be calibrated and determined offline. However, in practical usage, the diffusion model can be inferred with different numbers of timesteps and different solvers. Whether the timestep-wise quantization parameters

can generalize to unseen solvers or numbers of sampling steps remains challenging. (3) **Dynamic quantization acts as the algorithm performance upper bound for solving timestep-wise quantization:** The primary goal of timestep-wise quantization parameters is to reduce the quantization error caused by sharing the same set of quantization parameters across different timesteps. However, when employing dynamic quantization, no sharing of quantization parameters is involved, and such error is minimized. (4) **Comparison of hardware overhead:** A potential downside of dynamic quantization is the overhead involved in the online calculation of quantization parameters compared to static schemes that determine these parameters offline. As seen in Table, we have implemented efficient CUDA kernels, demonstrating that such overhead is acceptable (1.74x to 1.71x), while significantly improving algorithm performance by introducing dynamic quantization.

### E.2 HARDWARE IMPLEMENTATION OF QUANTIZED LINEAR LAYERS.

We present the process of quantized GEMM (General Matrix to Matrix Multiplication) in linear layers. It involves the following steps. Given a weights matrix of shape $[C_{in}, C_{out}]$ and an activation matrix of shape $[N_{token}, C_{in}]$, the matrix multiplication process can be described in the Figure Fig. 19. As can be seen, the elements for each row of the weight matrix and each column of the activation matrix need to be summed together. These values should share the same quantization parameter for efficient processing (so that the process of "integer computation with summation $W_{int}X_{int}$" and "multiplying by the quantization parameters $s_w s_x$" can be conducted separately). Therefore, the weight matrix should have "output-channel-wise" quantization parameters, and the activation matrix should have "channel-wise" quantization parameters.

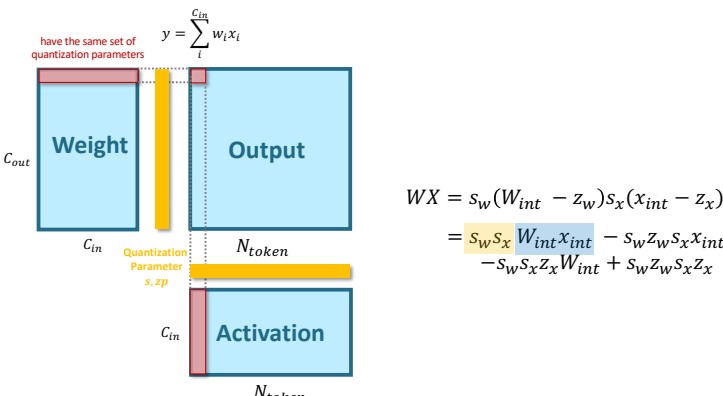

Figure 19: **Hardware implementation of quantized linear layer computation.** The yellow bar stands for the quantization parameters $s, z$.

### E.3 ANALYSIS OF METRIC DECOUPLED ANALYSIS

#### E.3.1 MOTIVATION FOR METRIC DECOUPLE

We verify the motivation of metric decoupled analysis through analyzing the layer type's correlation with the metric values. We compare and present each layer's sensitivity with respect to the metric value of each aspect in Fig. 20. The values are calculated as follows: Firstly, we calculate the relative metric value difference $(\text{Metrics}_{FP} - \text{Metric}_Q)/\text{Metric}_{FP}$. Then, we perform Z-score standardization for all values to ensure their values are within range of [0,1]. Next, we apply softmax to make each layer type's effect on different metrics sum to one. As can be seen, each layer type shows significant correlation with a certain metric, which corresponds to the model design. For instance, cross-attention, which is conducted between pixel and text embeddings, affects text-video alignment, and temporal attention, which models aggregation across frames, primarily affects the temporal consistency-related metric FlowScore. Despite some layer type (SeldfAttn) are sensitive to multiple metrics simultaneously, all layer types have a major focus on some aspects. The layer type that best fits the "sensitive to multiple metrics" category is the self-attention layers, which still predominantly

affect visual quality (0.6232) compared to other aspects (0.2364 and 0.1404). This still supports the motivation of adopting the metric decoupling approach.

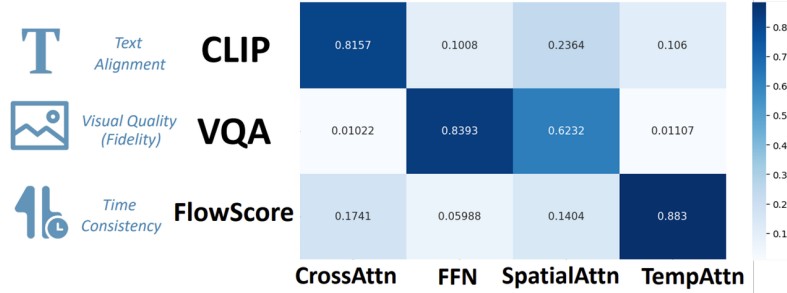

Figure 20: **Visualization of correlation between layer types and metric values.**

### E.3.2 DETAILED PROCESS OF METRIC DECOUPLED ANALYSIS

The metric-decoupled sensitivity analysis we introduced is a generalized framework that can be adapted to different tasks and models. It consists of three steps.

(1) Firstly, for a specific task (especially generative tasks), we need to identify common aspects that are typically evaluated and select corresponding metrics for them. These metrics can be chosen based on popular evaluation settings. For example, for video generation, we might consider visual quality (VQA), temporal consistency (FlowScore), and text-video alignment (CLIPSIM); for image generation, we might consider fidelity (FID) and text-image alignment (CLIP-Score).

(2) Next, we need to analyze how each part of the network affects these metric values. Given the large number of layers, we can group similar layers and conduct group-wise evaluation (for instance, grouping layers by operator types such as self-attention, cross-attention, and temporal attention). We can plot a heatmap (as seen in Appendix Section) to discover the correlation between layers and certain aspects. This correlation is intrinsically linked because of how the model is designed (for example, cross-attention strongly correlates with text-video alignment, as it is designed to model the correlation between text and image embeddings, and similarly for temporal attention). In addition to video generation, we observe similar phenomena in text-to-image generation with Pixart, where cross-attention layers primarily correlate with image-text alignment, and for self-attention and FFN layers with quality. Such findings could help us design "how to decouple the metrics".

(3) Finally, we need to "decouple" the effect on each layer type to obtain relative importance for more accurate sensitivity. With the guidance of the correlation from the previous step, we measure the relative importance of certain metrics (for instance, comparing all CrossAttn layers with their relative effect on CLIPSIM as sensitivity). The advantage of the "decouple" is two-folder. (1) It reduces the vast search space of jointly searching for each layer (by comparing only within groups). (2) It resolves the issue of different metrics' absolute value changes not being directly comparable. By carefully selecting the mixed precision plan for each group, it helps preserve multi-aspect metrics, avoiding the search from over-emphasizing certain aspects and causing failures in others. As presented in the results below and the qualitative results in Appendix, incorporating metric-decoupled analysis instead of joint search allows the generated quality to preserve multi-aspect metrics simultaneously.

### E.3.3 COMPARISON OF DIFFERENT MIXED PRECISION SEARCH METHOD.

We compare the "MSE-based", "Multiple metrics joint search based" mixed precision sensitivity analysis with metric decoupled analysis to demonstrate the effectiveness of metric decoupled mixed precision. As could be seen in Tab. 8, both the "MSE-based" and "Multiple metrics joint search based" achieves even worse results than uniform W4A8. We conclude the potential reasons for their failures as follows: For "MSE-based" analysis, as we discussed in Sec. 4.3, the MSE error could not accurately depict the changes in multiple aspects for video generation task. For "Multiple metrics joint search based", firstly, balancing the effect of different metrics is a non-trivial problem. Due to the diverse forms of various metrics, the absolute values of their changes cannot be directly

| Method | CLIPSIM | CLIP-Temp | VQA-A | VQA-T | $\Delta$ Flow Score |
|---|---|---|---|---|---|
| Without Mixed Precision | 0.181 | 0.999 | 60.216 | 42.257 | 0.151 |
| MSE-based Search | 0.179 | 0.999 | 53.335 | 38.729 | 0.258 |
| Multiple Metrics Joint Search | 0.179 | 0.999 | 51.256 | 35.412 | 0.279 |
| Metric Decoupled Search | 0.199 | 0.999 | 60.616 | 49.383 | 0.334 |

Table 8: **Comparison of different mixed precision analysis schemes under W4A8.**

compared. Therefore, we introduce metric decoupled analysis to ensure that each layer's sensitivity is measured with comparing with layers that have similar effects. To a certain extent, this approach demonstrates the principles of "controlled variable analysis".

