# OpenReview forum: "ViDiT-Q: Efficient and Accurate Quantization of Diffusion Transformers for Image and Video Generation"
_ICLR.cc/2025/Conference — ICLR 2025 Poster_

### Official Review · Reviewer_GGBr · 2024-11-04

**Soundness:** 3
**Presentation:** 3
**Contribution:** 2
**Rating:** 6
**Confidence:** 5

**Summary:**

This paper introduces ViDiT-Q (Video & Image Diffusion Transformer Quantization), a quantization scheme designed for Diffusion Transformers (DiTs) to reduce memory and computational demands in visual generation tasks like text-to-image and video synthesis. To address the unique challenges of DiT quantization, such as large data variation and time-varying channel imbalance, ViDiT-Q introduces fine-grained dynamic quantization for timestep-specific adjustments, a static-dynamic channel balancing technique, and a metric-decoupled mixed precision approach that allocates bit-widths based on layer sensitivity to visual quality metrics. Experiments demonstrate that ViDiT-Q achieves substantial hardware efficiency gains—up to 2-2.5x in memory savings and 1.4-1.7x in latency reduction—while maintaining high visual quality, making it a viable solution for deploying DiTs on constrained devices.

**Strengths:**

1. The paper introduces a unique approach tailored for Diffusion Transformers, addressing specific challenges like time-varying channel imbalance and data variation, which are rarely explored in quantization research.

2.ViDiT-Q’s methodology is grounded in thorough quantization error analysis, with each technique validated through extensive experiments on both text-to-image and text-to-video tasks, showing careful consideration of practical performance.

3. By achieving substantial memory and latency reductions, ViDiT-Q makes deploying DiTs on constrained devices feasible, enabling real-world applications for visual generation in resource-limited environments.

4.  The paper is well-structured and clearly explains each component of ViDiT-Q, supported by effective figures and diagrams that make the methodology and results accessible and easy to follow.

**Weaknesses:**

1. While the paper addresses W4A8 quantization, there is limited exploration of even lower bit-width configurations, such as W4A4 or W2A8, which are often critical for more aggressive compression on edge devices. A deeper analysis of these configurations would broaden the applicability of ViDiT-Q.

2. Although ViDiT-Q integrates several techniques, the paper lacks ablation studies that isolate the impact of each component, such as fine-grained dynamic quantization and static-dynamic channel balancing. Detailed ablations would provide clarity on the individual benefits of these methods.

3. The paper primarily compares ViDiT-Q to diffusion-specific quantization methods but lacks benchmarks against general quantization techniques (e.g., adaptive quantization). Including these comparisons would better contextualize ViDiT-Q’s performance.

4. The evaluation is restricted to text-to-image and text-to-video tasks, but the applicability of ViDiT-Q to other DiT applications (e.g., super-resolution or other image manipulation tasks) remains unexplored. Testing ViDiT-Q on these tasks could expand its impact and highlight potential limitations.

5. The hardware efficiency results are limited to a single hardware setup (NVIDIA A100). Testing on additional platforms, particularly lower-power devices, would provide more comprehensive insights into ViDiT-Q’s practicality for diverse deployment environments.

**Questions:**

1. Could you explore or discuss the feasibility of further quantizing ViDiT-Q to configurations like W4A4 or W2A8? Understanding its performance at lower bit-widths would clarify its limitations and potential for more aggressive compression.

2. Could you provide an ablation study showing the individual contributions of fine-grained dynamic quantization, static-dynamic channel balancing, and metric-decoupled mixed precision?

3. Have you considered benchmarking ViDiT-Q against broader quantization methods, such as adaptive quantization or knowledge distillation for compression?

4. In your metric-decoupled mixed precision approach, did you observe any limitations or challenges with determining sensitivity dynamically? Further details on how this sensitivity analysis adapts to different models or layers could provide practical guidance for real-world implementation.

---

### Official Review · Reviewer_9VJP · 2024-11-04

**Soundness:** 3
**Presentation:** 3
**Contribution:** 3
**Rating:** 6
**Confidence:** 2

**Summary:**

This paper proposes a quantization method tailored for diffusion Transformer models used in image and video generation. The authors conduct a detailed analysis of data scale variations across different components of the model, leading to the design of fine-grained grouping, dynamic quantization, and static-dynamic channel balance. To address the issue that quantization errors do not accurately reflect the quality of generation, they also introduce a metric-decoupled mixed-precision design. Experimental results show that the proposed method effectively improves the quality of generated outputs after quantization.

**Strengths:**

(1) The paper provides a detailed analysis of the sources of degradation in generation quality post-quantization and then proposes corresponding solutions. The article is easy to understand and is written with a clear logical flow.

(2) Experimental results demonstrate that, compared to existing methods, the proposed approach shows a significant improvement in generation quality.

**Weaknesses:**

The article is written with a relatively clear logic and has a certain degree of innovation. However, some parts still require further elaboration. Both the fine-grained grouping and dynamic quantization strategies are closely linked to existing methods. Yet, the author only briefly describes the differences without providing detailed, intuitive, or quantitative explanations. For instance, the specific distinctions between "channel-wise" and "output-channel-wise" are not clearly articulated, nor is it explained why "timestep-wise quantization parameters" would be more costly compared to the method proposed in this paper.

**Questions:**

Some questions have already been pointed out in Weakness.

Here, I have an additional question regarding the metric decoupled mixed-precision design. The paper emphasizes that different parts of the model affect generation quality in different ways. I would like to know to what extent these sensitivities are decoupled, and whether there are any modules that are sensitive to multiple metrics simultaneously. If such modules exist, should a joint sensitivity analysis involving multiple metrics be conducted?

---

### Official Review · Reviewer_2N42 · 2024-11-04

**Soundness:** 3
**Presentation:** 3
**Contribution:** 3
**Rating:** 6
**Confidence:** 2

**Summary:**

This paper introduces ViDiT-Q, a novel quantization method designed to address the unique challenges faced by diffusion transformers (DiTs) in text-to-image and video generation tasks. Large model sizes and multi-frame processing in video generation pose significant computational and memory costs, making efficient deployment on edge devices challenging.
The authors propose ViDiT-Q, a tailored quantization method for DiTs. This scheme effectively manages quantization errors by addressing specific challenges such as data distribution variations and channel imbalance. ViDiT-Q uses channel balancing to reduce color deviations and dynamic quantization to handle temporal variations in video sequences.

**Strengths:**

ViDiT-Q reduces the incoherence of data distribution, thereby lowering quantization error, by combining scaling and rotation-based channel balancing methods. Specifically, the scaling method addresses the "static" channel imbalance at the initial denoising stage, while the rotation method handles the "dynamic" distribution variations over time.

ViDiT-Q uses channel balancing to reduce color deviations and dynamic quantization to handle temporal variations in video sequences.

ViDiT-Q is validated on various text-to-image and video generation models, demonstrating minimal degradation in visual quality and metrics even at W8A8 and W4A8 quantization levels.

Qualitative results show that ViDiT-Q maintains high image quality and text-image alignment, while naive PTQ methods produce highly blurred or noisy images.

**Weaknesses:**

**Please note that since I am not an expert in model quantization and do not have any background in this field, the weaknesses I provide may not be sufficient to reveal the shortcomings of the work.**

1. While ViDiT-Q performs well at W8A8 and W4A8 quantization levels, there is a noticeable performance drop at lower activation bit-widths (such as W4A4 or W4A2). This indicates that the current mixed precision design has room for improvement, especially in fully leveraging the acceleration potential of 4-bit weights.

2.ViDiT-Q introduces multiple quantization parameters (such as different $\alpha$ values) to handle data variations across different timesteps. This complex parameter management increases the model's complexity.

3. I believe the model can further introduce 8-bit Attention (SageAttention) to improve model efficiency, which has already been integrated into some video diffusion model libraries. I wonder if 8-bit attention mechanisms or some linear attention acceleration mechanisms can further improve your solution?

**Questions:**

**Please note that since I am not an expert in model quantization and do not have any background in this field, the weaknesses I provide may not be sufficient to reveal the shortcomings of the work.**

1. While ViDiT-Q performs well at W8A8 and W4A8 quantization levels, there is a noticeable performance drop at lower activation bit-widths (such as W4A4 or W4A2). This indicates that the current mixed precision design has room for improvement, especially in fully leveraging the acceleration potential of 4-bit weights.

2.ViDiT-Q introduces multiple quantization parameters (such as different $\alpha$ values) to handle data variations across different timesteps. This complex parameter management increases the model's complexity.

3. I believe the model can further introduce 8-bit Attention (SageAttention) to improve model efficiency, which has already been integrated into some video diffusion model libraries. I wonder if 8-bit attention mechanisms or some linear attention acceleration mechanisms can further improve your solution?

---

### Meta-Review · Area_Chair_Zc26 · 2024-12-22

**Metareview:**

Summary:

The paper presents a post-quantization method for Diffusion Transformer models for image and video generation. It provides a detailed analysis of the source of degradation and proposes a tailored quantization method, achieving W8A8 and W4A8 with negligible degradation in visual quality and metrics.

Strength:
- Provide a detailed analysis of the sources of degradation. The proposed method is well-motivated.
- very well-written paper.
- demonstrating minimal visual quality and metrics degradation even at W8A8 and W4A8 quantization levels.

Weakness:
- There were concerns about the lack of lower bit-widths quantization, ablation study, and comparisons with relevant techniques. But the rebuttal addressed them well.

Justification:
All three reviewers' concerns have been adequately addressed. The responses and additional experiments are very detailed, according to the reviewers. The AC reads the reviews and response, and agrees with the reviewers that this is a solid contribution. The AC thus recommends to accept.

**Additional Comments On Reviewer Discussion:**

The AC belives that there are three main initial concerns:

1) Lower Bitwidth experiments (by Reviewer 2N42 and Reviewer GGBr):

The authors provide additional experiments on W4A4, W2A8 using the Pixart-Sigma model and report the FID, CLIP score, and ImageReward metrics. Both reviewers are satisfied with the additional exploration.

2) Lack of ablation study (Reviewer GGBr)

The authors present the ablation studies in Table 2 of the main paper. Specifically, they validate the importance of quantization parameters, channel balance, and mixed precision.

3) Lack of comparison with general quantization methods.

The authors provided the results in Table 6 in Appendix D.5. The results show general quantization methods like AdaRound and Brecq have moderate performance drops while the proposed ViDiT-Q achieves comparable results with the FP16 baseline.

---

### Decision · Program_Chairs · 2025-01-22

Accept (Poster)